

# Hydrodynamic Porosity:
# A Paradigm Shift in Flow and Contaminant Transport Through Porous Media, Part I

August H. Young[1,2], Zbigniew J. Kabala[3]

[1] Duke Center for WaSH-AID, Durham, NC, 27701, USA
[2] Mechanical Engineering and Materials Science, Duke University, Durham, NC, 27710, USA
[3] Civil and Environmental Engineering, Duke University, Durham, NC, 27710, USA

*Correspondence to*: August H. Young (ahf12@duke.edu)

**Abstract**

Pore-scale flow velocity is an essential parameter in determining transport through porous media, but it is often miscalculated. Researchers use a static porosity value to relate volumetric or superficial velocities to pore-scale flow velocities. We know this modeling assumption to be an *oversimplification.* The variable fraction of porosity conducive to flow, what we define as *hydrodynamic porosity*, $\theta_{mobile}$, exhibits a quantifiable dependence on Reynolds number (i.e., pore-scale flow velocity) in the Laminar flow regime. This fact remains largely unacknowledged in the literature. In this work, we quantify the dependence of $\theta_{mobile}$ on Reynolds number via numerical flow simulation at the pore scale. We demonstrate that, for the chosen cavity geometries, $\theta_{mobile}$ decreases by as much as 42% over the Laminar flow regime. Moreover, $\theta_{mobile}$ exhibits an *exponential* dependence on Reynolds number. The fit quality is effectively perfect, with a coefficient of determination ($R^2$) of approximately 1 for each set of simulation data. Finally, we show that this exponential dependence can be easily solved for pore-scale flow velocity through use of only a few Picard iterations, even with an initial guess that is 10 orders of magnitude off. Not only is this relationship a more accurate definition of pore-scale flow velocity, but it is also a necessary modeling improvement that can be easily implemented.

***Keywords:*** *Hydrodynamic Porosity, Cavity, Dead-End Pore, Pore Velocity, Volumetric Velocity, Reynolds Number, Groundwater Remediation*

## 1 Introduction

Groundwater is a primary drinking water source for 38% and 50% of the domestic and global populations, respectively (The Nature Conservancy, 2022). Despite mankind's undeniable dependence on clean groundwater, we have failed to properly protect it from anthropogenic pollutants. In the United States, the EPA Superfund Program oversees 1,336 sites on the agency's National Priorities List; another 40 sites are proposed to be added to this list (United States Environmental Protection Agency, 2023). Of the additional sites regulated by the Office of Environmental Management, there exists a total of 6.5 trillion liters of



contaminated groundwater (equaling roughly 4 times the country's daily water consumption) and 40 million cubic meters of contaminated soil (Office of Environmental Management, 2023a). The organization will spend 529 million USD remediating that volume of contaminated soil and groundwater in 2024 (Office of Environmental Management, 2023b).

Groundwater pollution is not just a pressing issue in the United States. For example, the Chinese State Council recently announced a groundwater remediation initiative equivalent to 5.5 billion USD. Though costly, the price for inaction is even higher: with 70% of the population dependent on groundwater, and 90% of shallow sources polluted (37% of which to the extent that they cannot be restored to drinking water quality, according to the Chinese Ministry of Land and Resources), 190 million Chinese residents fall ill every year, and 60,000 die, directly, or indirectly from groundwater contamination (Qiu, 2011).


The national groundwater cleanup debt for the United States was last estimated three decades ago at as much as 1 trillion USD by Russell et al. (1991) and highlighted by the National Resource Council (NRC). Adjusting for inflation, that's approximately *2 trillion USD* in groundwater remediation debt today. We believe that a comprehensive, updated estimate would be larger – *perhaps by an order of magnitude*. At the time of the previous estimate, the NRC also emphasized that the rate of discovery
of contaminated sites "far exceeded" the rate of cleanup (Mccarty, 1990). Even more notably, these cost estimates were made before discoveries of the contaminants that have emerged over the last 25 – 30 years and those that have resurfaced due to seemingly boundless prior use. Examples of such contaminants include fluorinated chemicals, known as PFAS, agricultural products, and cleaners. But these are just a few – researchers estimate that a total of eight million synthetic and naturally occurring chemicals were used in industrial, commercial, agricultural, and military activities over the past two centuries
(National Research Council, 2000). Further counteracting remediation efforts are phenomena that slow remediation efforts and result in contaminant rebound, e.g., DNAPL and matrix diffusion (Hadley and Newell, 2012). Still, other contributors arise from sources of geological contaminants (e.g., lead, arsenic, and other heavy metals), inadequate modeling of contaminant plumes, funding caps, and, of course, our inability to stop polluting. Altogether, these phenomena elongate remediation timeframes, and as a result, increase the associated cost.


With the bleak picture of our groundwater cleanup debt seemingly growing bleaker every year, it is unsurprising that practitioners, as well as government and funding agencies, increasingly embrace monitored natural attenuation (i.e., natural processes such as biodegradation, sorption, dilution, dispersion, and other chemical reactions that decrease contaminant concentrations) as an alternative to active treatment methods like pump-and-treat remediation. When conditions are favorable,
monitored natural attenuation should be considered as an alternative to energetically and economically expensive active remediation schemes. But these conditions are rare, and monitored natural attenuation is only appropriate for contaminants and sites that can be properly characterized over time.





Despite the shortcomings of natural attenuation, there has been a paradigm shift in the approach to groundwater remediation

– from active strategies (i.e., pump-and-treat) toward passive approaches (i.e., monitored natural attenuation). The study and

implementation of monitored natural attenuation grew rapidly in popularity since the 1990s; the same cannot be said for pump-

and-treat research, which remained relatively stagnant (see Figure 1). Evidence of this paradigm shift is materialized in

publications that provide guidance for the natural attenuation of PFAS, popularly known as "forever chemicals" (Newell et

al., 2021), and contaminants that are recognizably difficult to treat via natural attenuation (e.g., gasoline additives such as

methyl tert-butyl ether) (National Research Council, 2000). By 2000, The National Academies Press noted that natural

attenuation was approved "more and more frequently" despite a limited scientific understanding of the processes involved, the

potential risks, and the fact that it does not utilize continuously operated and supervised cleanup systems (National Research

Council, 2000).

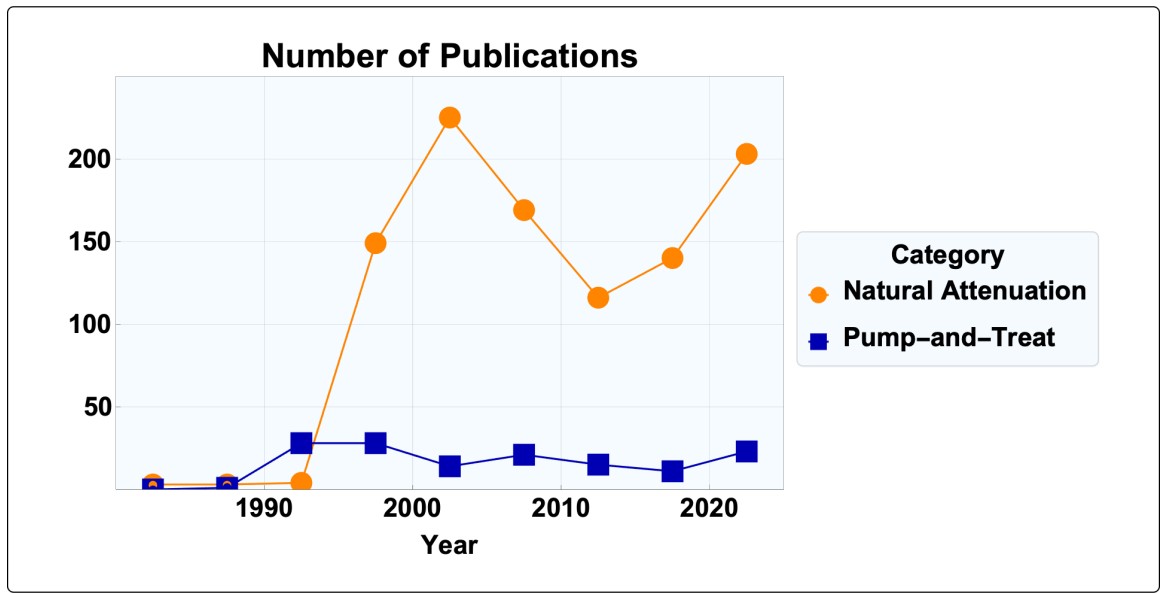


**Figure 1: Number of publications produced by title searches for "natural attenuation" (blue) and "pump-and-treat" (yellow) indexed by Web of Science in 5-year bins (1980-1984, 1985-1989, …, 2015-2019, 2020-2024); title search results are for February 28, 2023, with the contents of the last bin extrapolated from 2020 to this date.**

Because the definition of natural attenuation includes the dispersion and dilution of contaminants (National Research Council,

2000), it becomes an attractive, low-effort, low-cost alternative to active remediation strategies. The National Academies Press

highlights this reality, citing Arulanantham (1998): "Investigators and regulators sometimes employ and advocate minimalist

criteria or rules of thumb to make quick decisions on natural attenuation without using detailed technical protocols to show

cause and effect." Pressure comes from site owners, who know that the cost savings of natural attenuation can be in the millions

(USD) (National Research Council, 2000). Powers (1996) found that natural attenuation was 12 million USD less expensive

than a conventional pump-and-treat system at the French Limited site in Houston. But this is just for one site, suggesting far





more substantial savings for national corporations and agencies responsible for multiple site locations.Regrettably, monitored natural attenuation is usually a spurious alternative to pump-and-treat remediation, offering polluters the opportunity to skirt their cleanup responsibilities. Resulting, is the passage of our current groundwater remediation debt onto future generations.

In this work, we intend to reverse the observed paradigm shift in groundwater remediation strategy and resurrect the utilization of pump-and-treat systems. We justify this call by highlighting the fact that contaminated groundwater resources can be remediated at a *fraction* of formerly estimated costs. This is because remediation timeframes for pump-and-treat systems were previously estimated for *steady flows* and a *static definition of porosity* – a definition based on the mainstream, but outdated, single-porosity, double-porosity, and Mobile-Immobile Zone porosity models. The modeling of hydrodynamic porosity,

$\theta_{mobile}$, as a flow-dependent parameter is a *paradigm shift* in the understanding of flow and transport in porous media. The outdated Mobile-Immobile Zone model (wherein $\theta_{mobile}$ is assumed to be constant) is still in popular use today (e.g., consider the recent publications in leading journals: *Water Resources Research* (Li et al., 2022), *Environmental Science & Technology* (Zhou et al., 2023), *Journal of Hydrology* (Kwaw et al., 2023), *Groundwater* (Toyama et al.), etc.). The result is a poor presentation of induced subsurface flows and an overestimation of remediation costs.


Kahler and Kabala recently highlighted these potential cost-savings by demonstrating that remediation timeframes can be reduced by close to an *order of magnitude* for unwashed media (e.g., fractured rock and glacial deposits) subject to a rapidly pulsed flow; first via numerical simulation (Kahler and Kabala, 2016), then by physical experimentation (Kahler and Kabala, 2019). These results apply to contaminant removal from, as well as bacteria and nutrient delivery to, the dead-ended pore

spaces in unwashed media. And they offer hope that the national groundwater cleanup debt could be cut by almost an order of magnitude (a factor of seven, to be exact). For example, if we estimate that 30% of our contaminated groundwater formations are primarily unwashed media and assume that these formations would otherwise be cleaned by steady pump-and-treat, we would be looking at a cost savings of approximately 100 billion USD if we were to instead use rapidly pulsed pump-and-treat.

To utilize the cleanup process discovered by Kahler and Kabala (2016) (i.e., the deep sweeps and vortex ejections that manifest themselves only in rapidly pulsed pumping), we introduce a new concept, *hydrodynamic porosity*, $\theta_{mobile}$, to refer to the variable fraction of porosity used to transmit fluid through porous media. Without doing so, we cannot properly model the base flow that underlies the pulsations. For example, the predominant operational mode of pump-and-treat utilizes a circulation well that results in a dipole flow field, e.g., see Dinkel et al. (2020); Xia et al. (2019); Thomson et al. ( 2008); Sutton et al.

(2000); Kabala (1993); Kabala and Xiang (1992). In this flow pattern, velocities (and therefore Reynolds numbers) vary by orders of magnitude along flow streamlines. We illustrate this variation in Figure 2 below, by using the velocity field quantified by Philip and Walter (1992) for a confined, isotropic aquifer of infinite extent. For the highlighted streamline, we observe a four-order-of-magnitude change in the normalized pore-scale flow velocity. As qualitatively noted by Li et al. (1996) and Kabala and Kim (2011), changes in flow velocity result in changes to the volume of void space that is conducive to flow.





*Missing from the scientific literature is the explicit relationship between hydrodynamic porosity, $\theta_{mobile}$, and pore-scale flow velocity (or interstitial Reynolds number).*

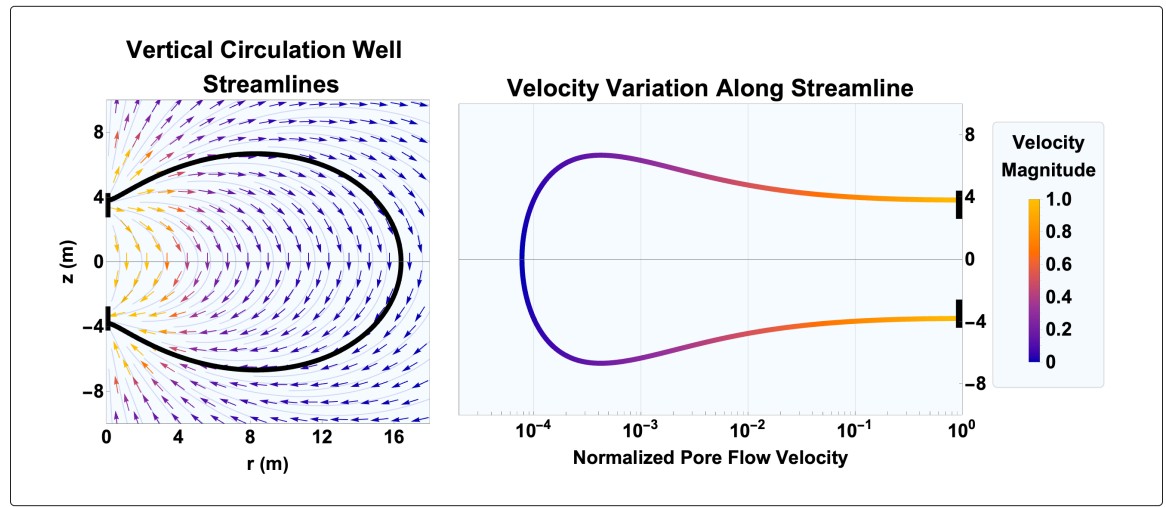

**Figure 2: Vertical circulation well streamlines for a confined, isotropic aquifer, mathematically described by Philip and Walter (1992) (left). Normalized velocity variation along the vertical coordinate of the streamline highlighted in black (right).**

Addressing this gap in the literature requires that we quantify and establish a relationship between hydrodynamic porosity, $\theta_{mobile}$, and flow velocity (Reynolds number). Given this relationship, researchers could determine the magnitude of the modeling errors that result when $\theta_{mobile}$ is replaced by a static parameter. The magnitude of these errors is especially consequential to decision-making calculations (i.e., those that pertain to the timescales and costs of remediation projects). In our subsequent work, we demonstrate the magnitude of the error that results when $\theta_{mobile}$ is replaced by a static parameter for
induced subsurface flows (such as the vertical circulation well we illustrate above).

## 2 Related Work

### 2.1 Effective Porosity

In this work, we introduce a new concept, *hydrodynamic porosity*, $\theta_{mobile}$, to refer to the variable portion of porosity able to transmit fluid through porous media. As we later demonstrate, $\theta_{mobile}$ varies with pore-scale flow velocity near sharp edges
and dead-end pores. We choose to use this term because it is not imbued with other, previously defined meanings. It differs from that of dynamic porosity, which describes the shrinking or swelling of porous media (Sheng et al., 2019; Mcdonald et al., 2020) dynamic effective porosity, which describes variably saturated porous media around the capillary fringe (Luo et al., 2023); and effective porosity, which is used by authors to describe a multitude of context-dependent phenomena.

Warranting further discussion is the concept and definition of effective porosity – a term commonly encountered in the study
of porous media flows. Researchers define effective porosity as the portion of constant porosity used to transmit fluid through





porous media. Although we could have used this work to describe the *hydrodynamic nature of effective porosity*, we refrain from using the term *effective porosity* due to its many alternative definitions that cloud the term in unnecessary ambiguity. Take, for example, the definition of effective porosity assigned by the textile industry. In the context of hernia meshes, effective porosity is meant to define changes to the pore morphology after implantation of the mesh in situ (Jacombs et al.,

2020). This is quite different from the definition used by Sevee (2010) in a study on the effective porosity of marine clay. In this study, effective porosity describes the void space in the clay that participates in advective transport. Still, other definitions describe it as the difference between the total porosity minus the soil water content at 0.33 bar (Helalia, 1993; Timlin et al., 1999). Readers are directed to Hapgood et al. (2002), Flint and Selker (2003), Cartwright et al. (2009), and Ma et al. (2022) for additional, alternative definitions. Given the highly varied use of effective porosity, we instead adopt the more precise

term, hydrodynamic porosity, $\theta_{mobile}$.

Among the authors that define effective porosity as the portion of constant porosity used to transmit fluid through porous media are Li et al. (1996), Kabala and Kim (2011), Kim (2006), Lindsay (1994), and Werth (2013). Here, we note that while these authors qualitatively consider its hydrodynamic nature, they do not quantify it. For example, Li et al. (1996) qualitatively

note the dependence of effective porosity on flow velocity in a study of sedimentary rock flows. The authors attribute the observed decrease to the presence of selective pathways (i.e., fissures and cracks) in the studied rock formation. They do not discuss an explicit relationship between what they term effective porosity and flow velocity. Further, the morphology of the rock formation studied in this work is starkly different than that of granular media.

In his doctoral dissertation at Duke, and a later publication with his advisor, Dr. Zbigniew Kabala, Kim (2006) and Kabala and Kim (2011) provide the most thorough discussion on the hydrodynamic quality of effective porosity; the authors state a dependence of effective porosity on both pore geometry and Reynolds number. They demonstrate this dependence in the same idealized pore space we study in this work using the FIDAP software. The authors show that effective porosity varies by at least an order of magnitude for creeping flows and postulate that it may "vary by as much" for non-creeping flows but do not

develop an explicit relationship between effective porosity and Reynolds number (Kabala and Kim, 2011).

Other authors have qualitatively considered the impact of dead-end pores on effective porosity. Lindsay (1994) notes that in water-saturated paper, flow can be restricted by mechanical obstructions in the form of isolated and dead-end pores. The author also discusses the development of stagnant zones in high Reynolds number flows, and the absence of these zones in creeping

flows due to the ability of the flow to closely follow abrupt changes in medium geometry. In effect, the author contemplates the dependence of effective porosity on Reynolds number but does not account for it in his analysis. The concept of flow-dependent porosity also appears in a study on baleen, though in a much different context. Werth (2013) describes a linear relationship between fringe porosity and incident flow velocity until a certain limiting velocity is reached, at which point the effective porosity of the baleen decreases. Given that the application of this work is to learn more about the mechanical





properties of baleen to better understand the feeding behaviors of various whale species, the spatial scale and flow path morphology are fundamentally different than those studied in this work.

## 2.2 Cavity Flows

As we discuss in the next section, flow past cavities (also known as dead-end or blind pores), which serve as the theoretical basis for this research, are a popular research topic and comparatively well-explored. There are ample publications concerning

contaminant transport and flow manipulation and instabilities in dead-end pores. The bulk of this work pertains to the geometric manipulation of pore geometry to determine the effect on the induced flow (Moffatt, 1963; Higdon, 1985; Shen and Floryan, 1985; Fang et al., 1997). Other studies on flow modulation have also been published, though this work is relatively less explored (Jana and Ottino, 1992; Howes and Shardlow, 1997; Horner et al., 2002; Kahler and Kabala, 2016). Most of this work is geared at industry for the guise of expediting rate-limited manufacturing processes (e.g., etching, finishing, cleaning,

etc.) that are applied to surfaces with cavities (Chilukuri and Middleman, 1984; Alkire et al., 1990; Fang et al., 1999).

In this work, we leverage the same physical theory that describes flows past cavities to define a hydrodynamic porosity function $\theta_{mobile}(\boldsymbol{v_{pore}}) = \theta_{mobile}(Re)$. Previously, Kahler and Kabala (2016, 2018, 2019), used the same approach to describe contaminant transport in porous media – likening duct or surface flow to flow through a series of well-connected pores, and

flow past grooves and cavities on surfaces to flow past dead-end pores in granular media. Such results are crucial to understanding how phenomena like contaminant rebound in groundwater reservoirs post-remediation can be mitigated. Although Kahler and Kabala implied the hydrodynamic nature of porosity, they did not quantify it. In this paper, we demonstrate and quantify, explicitly, this relationship for the first time. We also illustrate the ease with which this relationship can be incorporated into flow and contaminant transport models. To understand why contaminant rebound after traditional

pump-and-treat groundwater remediation takes place, and how it could be mitigated, researchers need to account for hydrodynamic porosity, $\theta_{mobile}$.

## 3 Physics

The idea of a medium having a singular porosity value is overly simplistic due to the many heterogeneities and time-dependent processes that occur in porous media. In larger structures, such as groundwater reservoirs, macrostructures and preferential

flow paths yield large and sporadic variations in porosity. In the case of packed beds, which are typically bounded, impermeable boundaries cause significant variations in porosity due to channeling effects at the wall. In rock formations, the concept of dual porosity is used to describe the very different characteristics of the medium itself and the fractures that separate the medium. Such morphologies require a definition of porosity with spatial dependence. Other pore-space reduction mechanisms like compaction and solute aggregation and deposition change the morphology of a medium over time. These

processes require a definition of porosity that is accommodating of these types of time-dependent processes. Both definitions,





however, are only dependent on the physical characteristics of the medium, whether over time or for an unchanging medium. What these definitions fail to account for, is the *nature* of the flow through porous media.

### 3.1 Pore-Scale Flow Velocity

Hydrodynamic porosity, $\theta_{mobile}$, is used to quantify transport through porous media. For example, because $\theta_{mobile}$ defines pore-scale flow velocity, and therefore the actual velocity field in porous media flows, Truex et al. (2017) utilize this parameter (although they utilize a static, effective porosity value) to calculate contaminant transport time. This is an improvement over the use of the volumetric velocity (also known as the superficial or Darcy/Forchheimer velocity), which is a hypothetical velocity equivalent to what would result from flow through an entire cross-section of the medium and not just its pore spaces:

$$\boldsymbol{v}_{volumetric} = \boldsymbol{v} = \frac{Q}{A} = \boldsymbol{q} \tag{1}$$

Where $Q$ is the volumetric flow rate, $A$, is the cross-sectional area of the medium, and $\boldsymbol{q}$, is the "flux." Neglecting inertial effects, Darcy's law relates the volumetric velocity to the pressure gradient applied to the medium (Brutsaert, 2005; Muljadi et al., 2016; Bear, 1975):

$$-\nabla p = \frac{\mu}{k}\, \boldsymbol{v} \Leftrightarrow -\nabla h = \frac{\mu}{k\, \gamma}\boldsymbol{v} \Leftrightarrow -\nabla h = \frac{1}{K}\boldsymbol{v} \tag{2}$$

Where $p$ is the pressure, $h$ is the pressure head, $k$ is the permeability of the medium, $K$ is the hydraulic conductivity, $\gamma$ is the specific weight of the fluid, and $\mu$ is its viscosity. The equivalences we show here are to illustrate the preferred forms in oil and gas reservoir modeling (left) and groundwater hydrology (right). When inertial effects cannot be neglected, as is the case for high Reynolds number flows, we must utilize the quadratic correction term, introduced by Èstudes (1863) and Forchheimer (1901). Given this adjustment, Darcy's law becomes the Forchheimer-Dupuit law:

$$-\nabla p = \frac{\mu}{k}\, \boldsymbol{v} + B\, \rho\, \boldsymbol{v}^2\, \boldsymbol{n} \;\Leftrightarrow\; -\nabla h = \frac{1}{K}\boldsymbol{v} + B\, \frac{1}{g}\, \boldsymbol{v}^2\, \boldsymbol{n} \tag{3}$$

where $\boldsymbol{n}$ is a unit vector in the direction of the volumetric velocity, $\rho$ the flow density, and $B$, a coefficient that can be found experimentally (Chen et al., 2015). Depending on the flow conditions, the volumetric velocity can be estimated experimentally from Eq. (2) or Eq. (3) (i.e., from Darcy's law or the Forchheimer-Dupuit law, respectively). To attain the true pore-scale flow velocity, which would be needed to determine quantities such as contaminant transport time, the volumetric velocity must be modified by the medium's porosity (Bear, 1975):

$$\boldsymbol{v}_{pore} = \frac{\boldsymbol{v}}{\theta} \tag{4}$$





In this formulation, $\theta$ is the effective porosity of the medium. Back-of-the-envelope calculations may simply use the total porosity of the medium. This would also be suitable for washed media without any cavities or other effectively immobile zones. Equation (4) can be derived from a simple conservation of mass analysis:

$$\boldsymbol{v}_{pore}\,A_{pore} = \boldsymbol{v}\,A \Leftrightarrow \boldsymbol{v}_{pore} = \frac{A}{A_{pore}}\,\boldsymbol{v} \Leftrightarrow \boldsymbol{v}_{pore} = \frac{1}{\theta}\,\boldsymbol{v}$$

Given our previous discussion, we know that use of the medium's total porosity is an oversimplification. Pore-scale flow velocity should instead be defined by the total volume that is conducive to flow (i.e., what we later refer to as mobile zones) – a quantity that is itself dependent on pore-scale flow velocity. Eq. (4) should instead read:

$$\boldsymbol{v}_{pore} = \frac{\boldsymbol{v}}{\theta_{mobile}\,(\boldsymbol{v}_{pore})} = \frac{\boldsymbol{v}}{\theta_{mobile}\,(Re)} \tag{5}$$

Where $\theta_{mobile}$ is itself a function of pore-scale flow velocity. The implicit nature of Eq. (5), while seemingly more difficult to
solve than Eq. (4), is quickly resolved by a few Picard iterations. Not only is Eq. (5) a more accurate description of pore-scale flow velocity, but it is also a necessary improvement in the modeling of induced subsurface flows that can be easily implemented.

### 3.2 Mobile-Immobile Zone Model

Porous media can be broken down into two regions: mobile and "immobile" zones, as described by Vangenuchten and
Wierenga (1976) in application to groundwater flows. In the mobile zone, solute transport occurs via advection and dispersion. The immobile zone is defined by isolated volumes of cavities or dead-ended pore space adjacently located to well-connected, mobile regions. In these zones, fluid recirculates in eddies, and solute transport is limited to the mechanism of vortex-enhanced diffusion. By this definition, the "immobile" label is a misnomer – fluid in the cavity space is *technically* mobile; it does not, however, move *through* the pore space. Thus, the fluid in this zone remains immobile *relative* to the flow in the mobile zone.

We illustrate these zones for an arbitrary matrix subject to an imposed flow in Figure 3, below. Magnification "A" provides an example of a mobile zone, and magnification "B" contains an example of an immobile zone in the form of a dead-end pore. The model we use to study this dead-end pore volume is discussed further in Section 3.5.




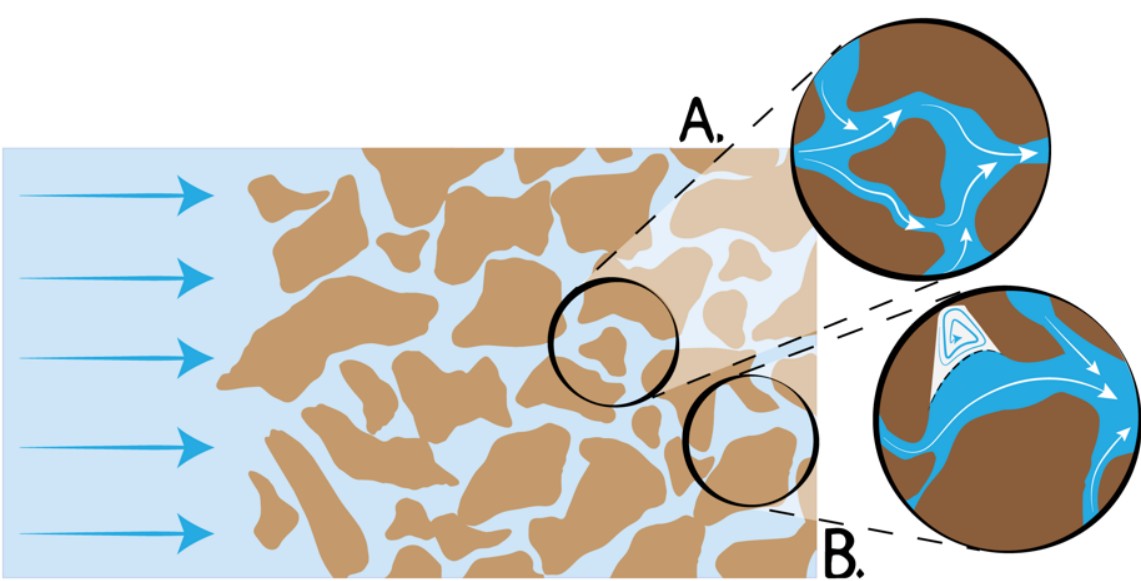

**Figure 3: Mobile zone composed of well-connected pore space (A), and an immobile zone in the form of a poorly connected/dead-end pore (B).**

### 3.3 Definitions of Porosity

The total porosity of a medium, $\theta$, is defined by the cumulative volume of the mobile and immobile zones; and more specifically, as the ratio of the total pore space volume to the total volume of the media. In an isotropic or 2D medium, like the

ones we study in this work, this definition can be written in terms of cross-sectional areas:

$$\theta = \frac{V_{pore}}{V} = \frac{A_{pore}}{A} \qquad (6)$$

We note that $V = V_{bulk} = V_{total}$ and $A = A_{total}$. In the analysis that follows, we represent the pore volume (or area) by the dead-end pore model, which we illustrate in Figure 4. As a result, the pore volume can be defined in terms of mobile and

immobile zones. Again, we provide the cross-sectional area expression for an isotropic medium or 2D media.

$$V_{pore} = V_{mobile} + V_{immobile} \rightarrow A_{pore} = A_{mobile} + A_{immobile} \qquad (7)$$

Thus, we can define the *hydrodynamic porosity* of the medium, $\theta_{mobile}$, by dividing Eq. (7) by the total volume of the medium, $V$ (or cross-sectional area, $A$):

$$\theta = \frac{V_{mobile} + V_{immobile}}{V} = \theta_{mobile} + \theta_{immobile} \qquad (8)$$





### 3.4 Defining the Pore-Space Partitioning Coefficient

Given that the pore space can be broken down into mobile and immobile zones, we can define a *pore-space partitioning coefficient*, $\xi$, to describe the ratio of pore space conducive to through-flow:

$$\xi \; = \; \frac{V_{mobile}}{V_{pore}} = \frac{A_{mobile}}{A_{pore}} \tag{9}$$

We note that the pore-space partitioning coefficient is related to the hydrodynamic porosity of the medium, $\theta_{mobile}$, by:

$$\xi \; = \; \frac{V_{mobile}}{V_{pore}} = \theta_{mobile} \, \frac{V}{V_{pore}} = \frac{\theta_{mobile}}{\theta} \tag{10}$$

In the analysis that follows, it is the *behavior of the pore-space partitioning coefficient* that we numerically quantify as a function of Reynolds number. We are able to use our results to describe the hydrodynamic porosity, $\theta_{mobile}$, of the medium because of the direct proportionality between these two quantities.

### 3.5 The Dead-End Pore Model

If we wished to geometrically simplify the pore space of a porous medium, we would study a single cavity or dead-end pore. The idea of a poorly connected, or dead-end pore was first explored by Turner (1958), who studied channel flow with distributed pockets of stagnant fluid. Although Turner admitted such pore spaces would play a role in diffusion throughout the pore space, other researchers such as Fatt (1959) and Goodknight et al. (1960) initially regarded dead-end pore spaces as regions through which diffusion could not occur. Deans (1963) noted that the division of pore space into flowing and stagnant regions, separated by a "resistance to mass transfer" is an "extreme limit" that can only be justified on the grounds of simplicity. Following this conclusion, Coats and Smith (1964) relaxed the definition of the dead-end pore to account for diffusion, but still referred to the dead-end pore volume as stagnant. Physically, we know this enforcement to be an oversimplification of the recirculatory flow within the dead-end pore space. Chilukuri and Middleman (1984) corrected for this oversimplification by describing mass transport from dead-end pores as a result of vortex-enhanced diffusion – a conclusion that coincides with a series of publications that detail the vortex structures within dead-ended pores (Moffatt, 1963; Mehta and Lavan, 1969; O' Brien, 1972; Shen and Floryan, 1985; Kang and Chang, 1982; Fang et al., 1997). The evolution of the dead-end pore model is illustrated in Figure 4, below.





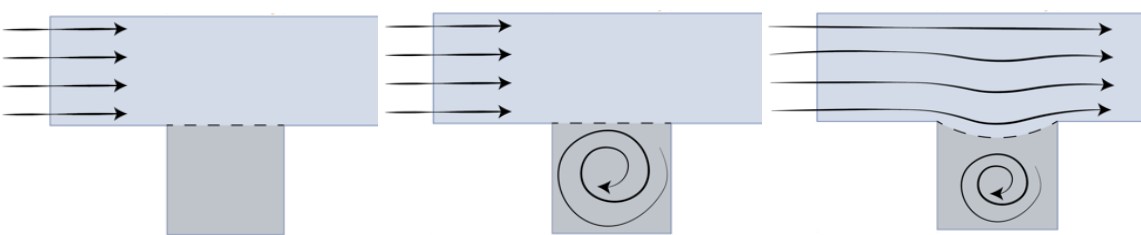

**Figure 4: Mobile-Immobile Zone model of the dead-end pore space (left), boundary-driven condition, and vortex-enhanced diffusion**
**(middle), shear-driven boundary condition yielding a deformable, mobile separatrix (right).**

Separation of the mobile and immobile zone volumes is described by the boundary- or shear-driven flow models; we refer to the boundary between these zones as the separatrix. In application to flow, the idea of a separatrix was first postured by Elderkin (1975), who describes the boundary as a trajectory that is topologically abnormal in comparison to nearby trajectories. Weiss (1991) later used this concept to describe the defining limit between free and trapped fluid regions. Other publications refer to this boundary as a dividing (Moffatt, 1963; O' Brien, 1972; Higdon, 1985) or separating streamline (Shen and Floryan, 1985; Alkire et al., 1990). The modeling and experimental work on which we build, i.e., Kahler and Kabala (2016, 2018, 2019), and even earlier publications such as Horner et al. (2002), use this same terminology (the separatrix) to describe the fluidic boundary between the mobile and immobile zones in the idealized dead-end pore space.

In the case of the boundary-driven model (which is essentially the commonly studied driven-lid problem), the geometric boundary between the mobile and immobile zones also serves as the fixed location of the separatrix. With an increase in Reynolds number of the adjacent through-channel flow, the vortex structures within a cavity translate and smear in the direction of the imposed boundary condition movement. Such results have been illustrated by many and summarized by Shankar and Deshpande (2000). In shear-driven flows, the separatrix is free to move about the cavity space. As discussed by Fang et al. (1999) and Kahler and Kabala (2016), the exact location of the separatrix depends on the Reynolds number of the adjacent through-channel flow in the mobile zone. This means that, unlike in the case of the boundary-driven flow condition, the mobile and immobile zones cannot be defined based on the geometry of the pore space alone. Instead, the volumes of these zones must be defined as flow dependent. In application to square cavity flow, comparison between the enforcement of the boundary-driven and shear-driven flow conditions is provided in Figure 5 below.





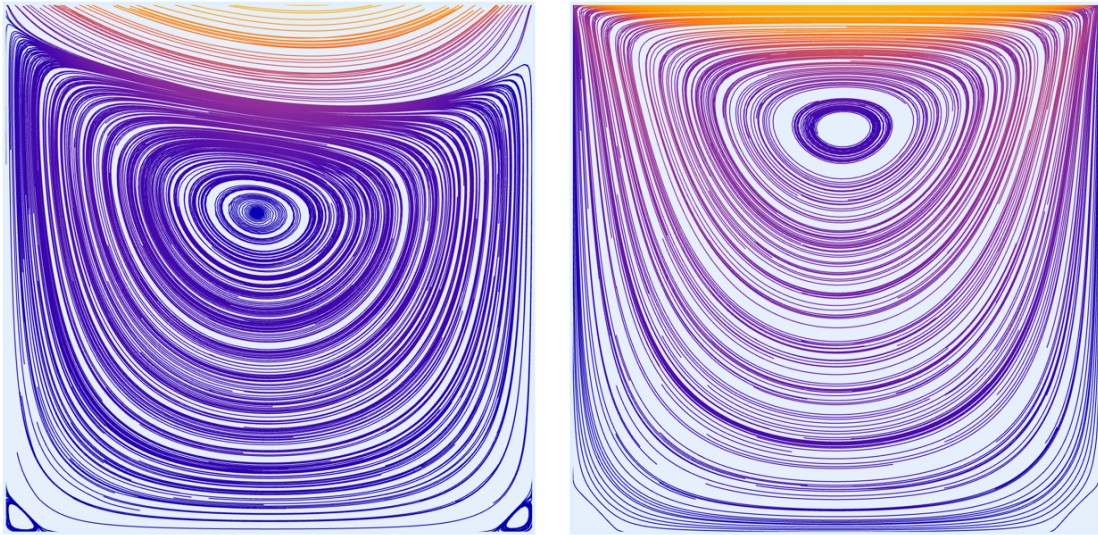


**Figure 5: Vortex location for the mobile and immobile separatrix (left and right, respectively). The location of the mobile separatrix is determined from a shear-driven flow condition. The location of the immobile separatrix is determined from a boundary-driven flow condition. These results are generated for the idealized pore space provided in Figure (4) for *Re* = 10.**

In the study of flow past cavities, it is standard practice to enforce the shear-driven boundary condition. Researchers that
initially studied these flows, e.g., Moffatt (1963), immediately identified through-flow penetration into the cavity space upon
investigation. O'neill (1977) and Wakiya (1975) found that attachment of the separatrix to the cavity wall occurs at some depth
into the cavity and as Higdon (1985) states, not at the sharp, leading edge of the cavity. In fact, for rectangular cavities
exceeding a given depth ratio, researchers found that the downstream attachment of the separatrix, or dividing streamline,
occurs at the bottom of the cavity wall, mimicking the behavior of a sudden-expansion flow (Shen and Floryan, 1985; Alkire
et al., 1990). Enforcement of the boundary-driven flow model would, in this application, yield significant error.

With these findings considered, we again refer to the discussion presented by, Li et al. (1996) wherein the effective porosity
of the studied medium is found to be flow dependent. The authors of this study explain that this dependence is a result of
physical macrostructures that act as preferential flow paths during high flow volumes. What the authors do not discuss, is the
hydrodynamic behavior of the immobile zones within the medium – a behavior that is driven by separatrix movement in to
and out of each effectively dead-end pore. As flow volumes increase, the separatrix moves toward its neighboring through-
channel, and the immobile zone it defines grows. The result of an increase in flow volume is a decrease in hydrodynamic
porosity, $\theta_{mobile}$. It is only when the shear-driven flow condition is applied to the dead-end pore that this behavior is observed.
If the mobile and immobile zones are improperly defined by the simplified boundary-driven flow condition, this behavior is
missed.





## 4 Methods

To determine the hydrodynamic porosity, $\theta_{mobile}$, of a porous medium, we study the medium at the pore scale and assume an idealized dead-end pore geometry. The movement of the separatrix is tracked over a range of interstitial Reynolds numbers to determine the relative magnitudes of the mobile and immobile zones, which we then use to calculate the value of the pore-space partitioning coefficient, $\xi = \theta_{mobile}/\theta$.

### 4.1 Numerical Flow Solver

To observe movement of the separatrix in the idealized pore space, we use *Mathematica*'s numerical differential equation solver, *NDSolve*, to solve the continuity equation (mass conservation), Navier-Stokes equations (momentum evolution), and associated boundary conditions. The solver domain is a replica of the idealized pore space utilized by Kahler and Kabala (2016) and is similar in geometry to the domain commonly used in the study of flow past cavities (Chilukuri and Middleman, 1984; Higdon, 1985; Fang et al., 1997). For the sake of simplicity, the flow is modeled as being two-dimensional. The height of the through-channel is the same as the depth and width of the dead-end pore (i.e., the dead-end pore has a depth ratio of 1:1). In the study of flow past cavities, this geometry is by far the most prevalent, as noted by Shankar and Deshpande (2000). The through-channel is extended past the dead-ended pore by twice the channel height to eliminate any end effects associated with the outflow boundary condition. Finally, the dead-ended pore is located one-fourth of the way into the through-channel given the need to input a fully developed flow profile at the through-channel inlet. To exaggerate the movement of the separatrix as a function of Reynolds number, the solver domain is manipulated such that the through-channel becomes much narrower than the depth and width of the cavity space.

The idealized flow geometry is discretized through use of the *ElementMesh* function, which, by default, generates a second-order, triangular element mesh. The interior and boundary mesh elements are further refined by specifying upper limits on the *MaxCellMeasure* and *MaxBoundaryCellMeasure*. A brief convergence analysis of the interior and boundary mesh cell sizes is provided in the supplemental material. Further, a refinement region is specified at the geometric boundary of the channel-cavity interface to ensure proper resolution of the separatrix.

The solver itself is defined by the system of equations that describe steady-state flow through the idealized pore space (i.e., the incompressible form of the continuity and Navier-Stokes equations), as well as the boundary conditions that are assigned to the solver domain. These equations are normalized by the channel height, $h$, and the average inlet flow velocity, $U$. The flow is assumed to be steady-state and restricted to the laminar flow regime. A set of Dirichlet conditions are applied to the boundaries of the solver domain (i.e., the no-slip condition at the domain walls and a uniform pressure condition, wherein the pressure is arbitrarily set to zero, at the domain outlet). The inlet velocity profile is defined by the Hagen-Poiseuille model for





fully developed channel flow. Finally, flow is assigned to the entire idealized pore space given that the application of this work is to fully saturated porous media. The properties of water at standard conditions are assigned to the fluid.

Here it is important to note that the non-dimensional form of the Navier Stokes equations is used in this analysis. The scaling on the pressure term is appropriate for flows that are dominated by convective action (i.e., flows in which viscous effects are relatively negligible). This choice was made to replicate the scaling utilized by Kahler and Kabala (2016), who studied flows with channel-based Reynolds numbers of 0.01–10. Fang et al. (1997) used the same scaling, though for admittedly higher channel-based Reynolds numbers in the range of 50–1,600. For our case, and the work on which we build, this scaling is justified because these studies aim to capture flow phenomena driven by convective action (i.e., changes in momentum to the bulk flow). When the Reynolds number approaches 0, and the flow can be approximated as creeping, the scaling on the pressure term can be achieved through use of the flow viscosity.

For this system of equations, *NDSolve* utilizes Finite Element Method to arrive at a solution. In general, the solver method utilized by *NDSolve* is automatically determined by the results of symbolic analysis. The use of Finite Element Method is triggered by specific user inputs. For example, specification of the Navier-Stokes equation using the 'Inactive' operator or boundary conditions defined by the *DirichletCondition* function prompt the use of this solver method. Implementation of this solver method can be verified by validating that the solution contains an *ElementMesh* (Wolfram, 2022). The outputs of the solver are three interpolating functions that describe the pressure and velocity fields within the flow domain. Streamlines are visualized through use of the built-in *StreamPlot* command. The error associated with each solver output is determined by the solver mesh (i.e., domain discretization), assigned working precision, and solver method.

### 4.2 Data Collection

In this work, we vary the Reynolds number of the through-channel flow (i.e., the flow in the mobile zone), and the depth ratio of the idealized dead-end channel-cavity geometry. As noted by Fang (2003), and other publications, the location of the separatrix is a function of both Reynolds number and geometry (Mehta and Lavan, 1969; O' Brien, 1972; Higdon, 1985; Kim, 2006).

### 4.2.1 Reynolds Numbers

Below a Reynolds number of 1, the separatrix remains stationary (Kahler and Kabala, 2016). Flows of this nature are classified by the creeping flow regime, where viscous effects dominate. It is not until we study Reynolds numbers within the inertial flow regime that we observe a mobile separatrix. This is because the location of the separatrix is dictated by the inertia of the adjacent through-flow. For this reason, we impose the following Reynolds number, *Re*, ranges to the through-flow:

- *Re* = 0.01 – 1

  (to verify the stationary nature of the separatrix in the creeping flow regime)





- *Re* = 1 – 100

  (to illustrate the mobility of the separatrix in the laminar flow regime)

To remain within the laminar flow regime, we limit our Reynolds number to a maximum of 100. This choice is admittedly arbitrary, given that transition to turbulence in pipe flow typically occurs over a diameter-based Reynolds number of 2,000 and at least one order of magnitude above the particle-based Reynolds number at which the deviation from Darcy's Law occurs in porous media (Bear, 1975); deviation from Darcy's Law generally occurs between a particle-based Reynolds number of 1 and 10.


In terms of particle-based Reynolds numbers, there is ample evidence that the onset of transitionary behavior occurs around 100. For columns of packed spheres, Jolls and Hanratty (1966) report the onset of transitionary behavior within the range of 110 – 150. Wegner et al. (1971) found a slightly lower range of 90 – 120 for beds of packed spheres. Latifi et al. (1989) encountered transitionary behavior at 110, also for a bed of packed spheres; however, the authors did note unsteady laminar

flow behavior until 370. A similar study conducted by Rode et al. (1994) reports transitionary behavior in the range of 110 – 150. Finally, Bu et al. (2014) define a critical particle-based Reynolds number of 100 as the cutoff for laminar flow, with the onset of turbulence occurring between 230 – 400.

If we instead consider the interstitial Reynolds number, which is based on average pore size and average pore-scale flow
velocity, we encounter the commonly cited Reynolds number ranges provided by Dybbs and Edwards (1984), summarized in Table 1 below.

**Table 1: Reynolds number ranges corresponding to pre-turbulent flow regimes, as provided by Dybbs and Edwards (1984).**

| Flow Regime | Reynolds Number Range |
|---|---|
| Creeping/Darcy | 0 – 1 |
| Inertial | 1 – 10 |
| Laminar, non-linear | 10 – 150 |
| Laminar, unsteady | 150 – 300 |

We can easily imagine representing our system in terms of the interstitial Reynolds number. Given our assumption that the
medium is homogeneous, we know the average pore size. Pore-scale flow velocity is typically calculated by dividing the flux through the medium by the porosity of the medium, but in our case, we will impose it directly by assigning a mean velocity to the through-channel flow. Capping our Reynolds number at a value of 100 keeps our analysis within the steady laminar flow regime.





### 4.2.2 Flow Geometries

The effect of pore geometry manipulation has been extensively studied in the literature. In these studies, the authors vary the type of cavity (i.e., rectangular, circular, etc.), the depth ratio of the cavity, and the size of the cavity relative to the size of the adjacent through-channel. To replicate the results obtained by Kahler and Kabala (2016), we use an idealized geometry wherein the height of the through-channel is equivalent to the depth and width of the cavity geometry, as provided in the top-left quadrant of Figure 6. To exaggerate the mobility of the separatrix in the laminar flow regime, the depth and width of the cavity,

relative to the through-channel, are equally increased in magnitude. These geometries are provided in Figure 6, below, and referred to by channel-cavity depth ratios (i.e., 1:1, 3:4, 1:2, and 1:4). For example, the depth ratio 1:2 corresponds to the geometry in which the cavity depth and width are twice that of the through-channel height.

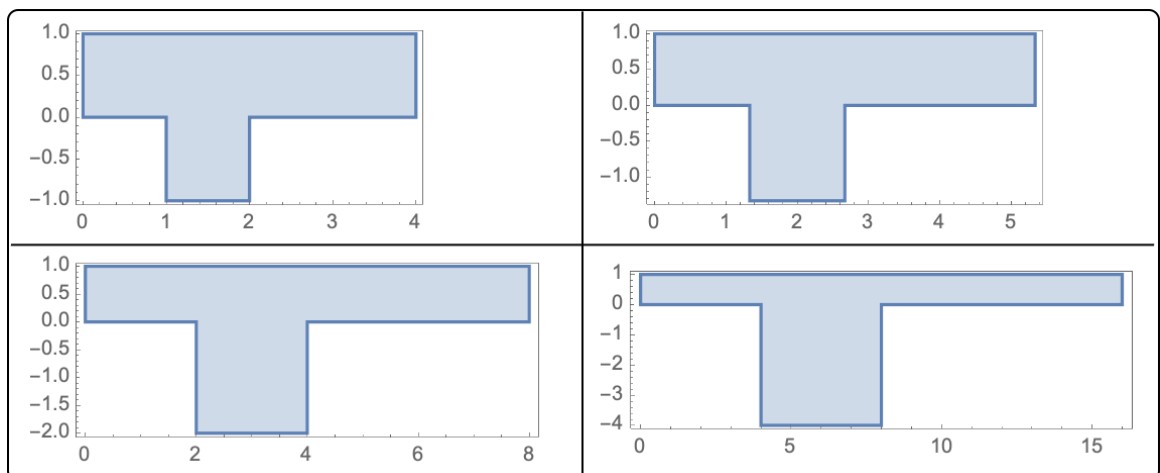

**Figure 6: Study geometries 1:1 3:4, 1:2, and 1:4; specified by their depth ratio (i.e., channel height to cavity depth/width) in non-**
**dimensional space.**

### 4.3 Measurement Method

To determine the value of the pore-space partitioning coefficient, $\xi = \theta_{mobile}/\theta$, and therefore the hydrodynamic porosity, $\theta_{mobile}$, for each inlet flow condition, we use the following procedure:

1. Generate a monochromatic stream plot of the flow,


2. Draw the separatrix in the area between the bulk flow in the through-channel and the recirculatory flow in the dead-end pore space in a contrasting color, using the streamlines in the stream plot for guidance,

3. Use an interpolating function to mathematically describe the location of the separatrix,

4. Define the area below the separatrix as the immobile zone, and the area above as the mobile zone and use numerical integration to quantify the magnitude of these regions.





To draw the separatrix, we use a *DynamicModule* in *Mathematica* to generate an interpolating function that includes five points (or more) of our choosing between the mobile and immobile zones. This process is pictured below in Figure 7. To determine the sizes of the mobile and immobile zones, we apply numerical integration to the resulting interpolating function.

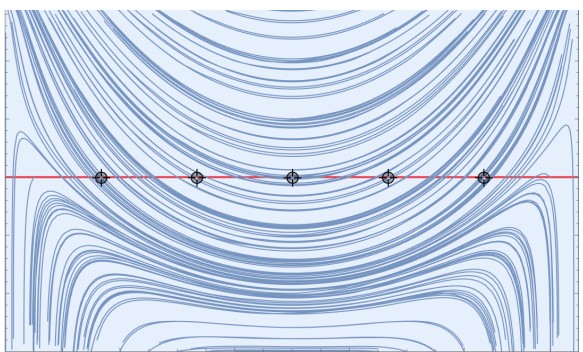 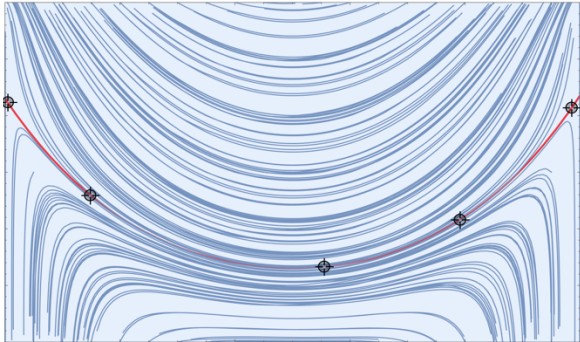

**Figure 7: A *DynamicModule* automatically places five points on a monochromatic stream plot in the vicinity of the separatrix(left).**
**Given user input (i.e., movement of these five points to the approximate location of the separatrix), the *DynamicModule* produces an interpolating function that can be used to describe the location of the separatrix (right).**

Given that the shape and location of this interpolating function are a direct result of user input, there is an inherent error built into the measurement process that we are unable to precisely quantify. Additional errors in the measurement process result from the chosen resolution of the stream plots which is in turn limited by the quality of the solver mesh and working precision
assigned to the numerical solver method.

## 5 Results

### 5.1 Separatrix Movement

Because the cavity flow is driven by the adjacent through-channel flow, we start by providing a stream plot of the entire dead-end pore geometry adjacent to the corresponding cavity flow in Figure 8. We then provide stream plots for each cavity
geometry at Reynolds numbers of 1, 10, 50, and 100 in Figure 9.

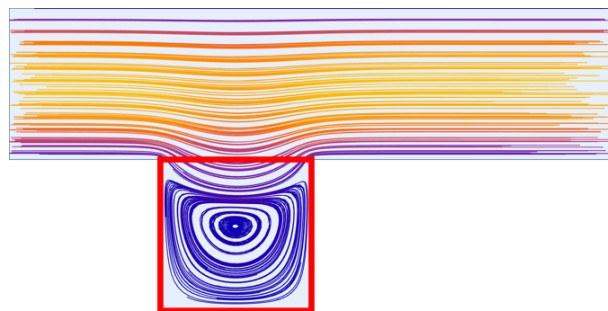 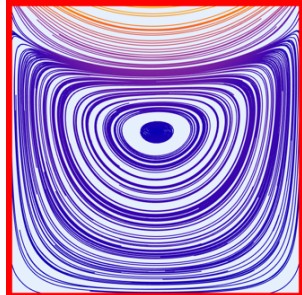

**Figure 8: Stream plot magnification example.**








**Figure 9: Dead-end pore stream plots for each depth ratio (1:1, 3:4, 1:2, and 1:4 from top to bottom) for Reynolds numbers (1, 10, 50, and 100 from left to right).**

Movement of the separatrix as a function of Reynolds number was first explored by Kahler and Kabala (2016). In their work,

the authors track the bottom-most point of the separatrix to determine its maximum penetration depth into the dead-end pore. To confirm the numerical accuracy of the results produced in this work, we replicate this plot, which is provided in the supplemental material. We expand upon this plot by tracking the movement of the separatrix for three additional flow geometries, as pictured below.





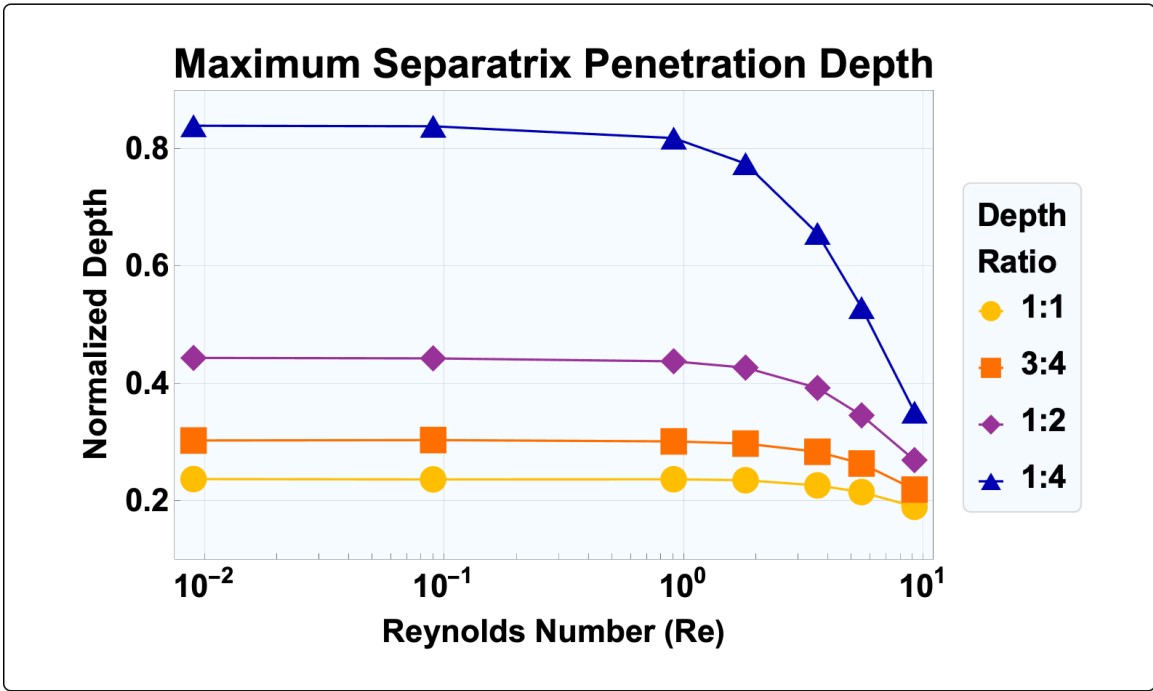

**Figure 10: Maximum relative penetration depth of the separatrix into the dead-end pore space as a function of Reynolds number, *Re*.**

The results obtained by Kahler and Kabala (2016) reveal an immobile separatrix in the creeping flow regime (i.e., *Re* < 1), and a mobile separatrix in the inertial flow regime. In the former, the bottom-most point of the separatrix does not exceed 25% of the depth of the idealized pore space. Our results replicate the separatrix behavior observed by Kahler and Kabala (2016) and are within, at most, a 2% difference. Over the range of Reynolds numbers plotted in Figure 10, the maximum penetration depth of the separatrix diminishes by 20%.

Building upon the test conditions utilized by Kahler and Kabala (2016), we observe movement of the separatrix toward the geometric boundary of the cavity space for Reynolds numbers approaching 100. Movement of the separatrix within the cavity space is further exaggerated by manipulation of the flow geometry, as illustrated in Figure 11 below.





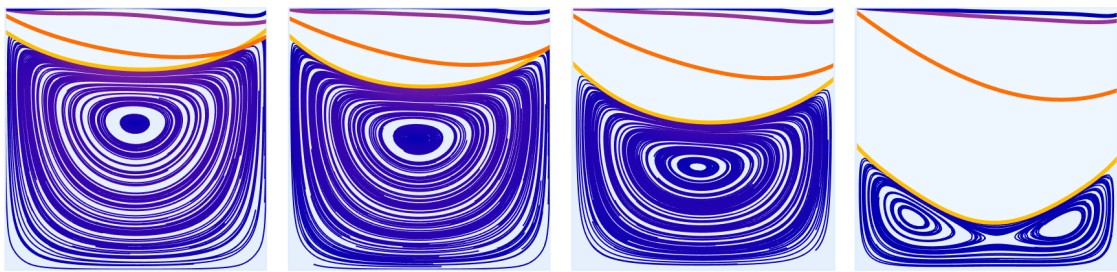

**Figure 11: Immobile zone vortex streamlines, corresponding to *Re* = 1, bounded by the separatrix (highlighted in yellow) for each channel-cavity depth ratio, 1:1, 3:4, 1:2, and 1:4 (from left to right). Additional separatrix locations for Reynolds numbers corresponding to 10, 50, and 100 are plotted in orange, purple, and blue, respectively.**

Movement of the separatrix toward the adjacent through-channel (and out of the cavity space) results in a decrease in the pore-space partitioning coefficient, $\xi = \theta_{mobile}/\theta$, and therefore the hydrodynamic porosity, $\theta_{mobile}$, of the medium. For example, media with cavities described by the dead-end pore model pictured in Figure 4 experience roughly a 4% reduction in $\theta_{mobile}$ over the tested Reynolds number range. By comparison, when the channel-cavity depth ratio is 1:4 (i.e., the cavity depth is 4 times that of the through-channel height), $\theta_{mobile}$ decreases by approximately 42%. See Table 2 for a summary of the changes

associated with each channel-cavity depth ratio; note that given Eq. (10), the provided percent-decrease values are the same for the pore-space partitioning coefficient, $\xi$, and the hydrodynamic porosity, $\theta_{mobile}$, of the medium.

**Table 2: Maximum and minimum pore-space partitioning coefficient, $\xi = \theta_{mobile}/\theta$, values corresponding to an increase in Reynolds number in each tested flow geometry.**

|  | Pore-Space Partitioning Coefficient, $\xi = \theta_{mobile}/\theta$ | | |
| --- | --- | --- | --- |
| **Depth Ratio** | **Max.** | **Min.** | **% Decrease** |
| 1:1 | 0.84 | 0.8 | 4.34 |
| 3:4 | 0.81 | 0.75 | 7.47 |
| 1:2 | 0.79 | 0.67 | 15.81 |
| 1:4 | 0.87 | 0.51 | 41.50 |

**5.2 Exponential Dependence of Hydrodynamic Porosity on Pore-Scale Flow Velocity**

When plotted, the pore-space partitioning coefficient, $\xi = \theta_{mobile}/\theta$, and therefore the hydrodynamic porosity, $\theta_{mobile}$, of the medium, approaches the value suggested by the boundary-driven model we previously discussed for Reynolds numbers approaching the upper limit of the laminar regime (*Re* = 100). See Figure 12 (below) for evidence of this behavior. In this figure, we also demonstrate that these quantities exhibit a linear dependence on Reynolds number, in the inertial flow regime

(*Re* = 1 – 10). Extrapolation of this relationship past *Re* = 10 is provided to illustrate the error that would result from not





utilizing the exponential relationship provided in Eq. (12). Again, we remind readers of the direct proportionality between the partitioning coefficient, $\xi = \theta_{mobile}/\theta$, and the hydrodynamic porosity of the medium, $\theta_{mobile}$.

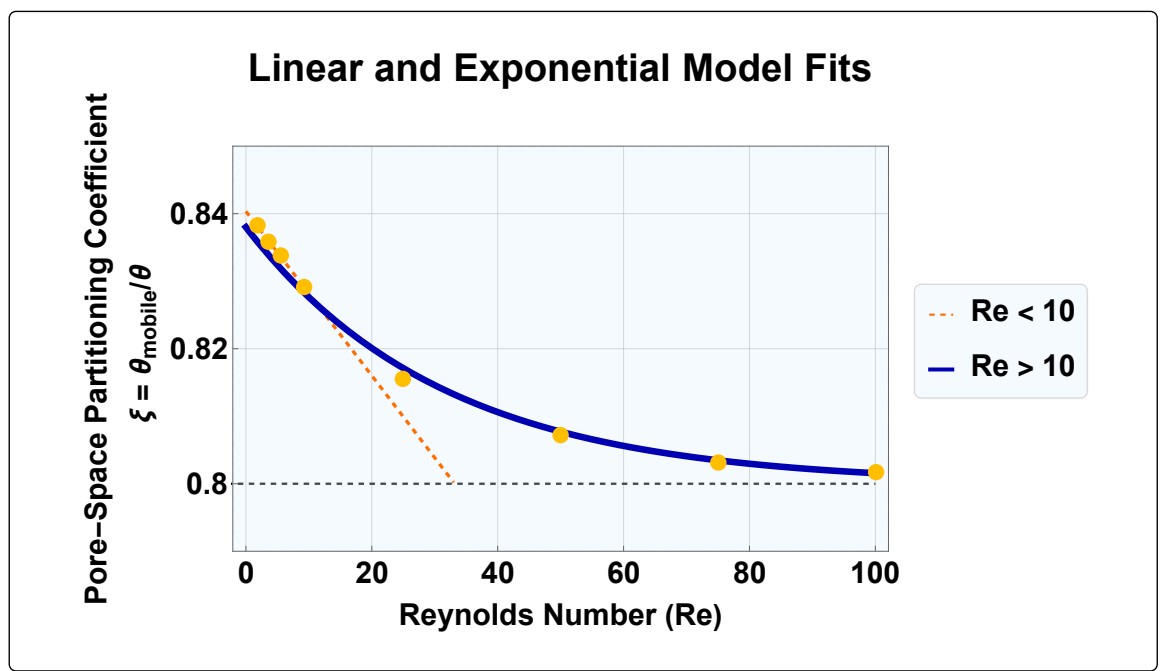

**Figure 12: The pore-space partitioning coefficient, $\xi = \theta_{mobile}/\theta$, exhibits a linear dependence on Reynolds number, *Re*, for *Re* = 1–10 and an exponential dependence for *Re* = 1–100. At Reynolds numbers approaching 100, the partitioning coefficient of the flow geometry approaches the value predicted by the boundary-driven flow model. The results pictured here are for the 1:1 channel-cavity depth ratio.**

The fit parameters for each channel-cavity depth ratio are provided in Table 3. The quality of each fit is measured by the coefficient of determination ($R^2$), which is approximately 1 for each tested depth ratio. In the inertial flow regime, the pore-space partitioning coefficient, $\xi = \theta_{mobile}/\theta$, exhibits a linear dependence on Reynolds number. However, as the Reynolds number increases, the partitioning coefficient deviates from this linear model. Instead, a nonlinear dependence on Reynolds number explains the calculated partitioning coefficient values at higher Reynolds numbers, and more significantly, over the *entire* range of Reynolds numbers, *Re*, in the laminar regime (*Re* = 1–100). This dependence is well-fit by an exponential function:

$$\xi = a + be^{-c\,v_{pore}} \iff \xi = a + be^{-d\,Re}, d = \frac{c\ height}{v} \tag{11}$$

This expression can be easily re-written using the help of Eq. (10) to define the hydrodynamic porosity of the medium, $\theta_{mobile}$:

$$\theta_{mobile} = (a + be^{-c\,v_{pore}})\,\theta \iff \theta_{mobile} = (a + be^{-d\,Re})\,\theta \tag{12}$$





Where $a$ is the *pore-space partitioning coefficient*, $\xi = \theta_{mobile}/\theta$, value approximated by the boundary-driven model (i.e., $Re \rightarrow \infty$), and the quantity '$a + b$' is the value in the creeping flow regime (i.e., $Re \rightarrow 0$). We note that these values, as written,
are the same for Eq. (11) and (12).

Table 3: Hydrodynamic porosity, $\theta_{mobile}$, parameters for Reynolds number dependence (d) and pore-flow velocity (c) defined in Eq. (11) and (12).

| Depth Ratio | Equation (12) Exponential Fit Parameters | | | | |
|---|---|---|---|---|---|
| | $R^2$ | $a$ | $b$ | $c$ (s/m) | $d$ |
| 1:1 | 1.0000 | 0.80 | $3.80 \times 10^{-2}$ | 25.91 | $3.19 \times 10^{-2}$ |
| 3:4 | 1.0000 | 0.75 | $6.20 \times 10^{-2}$ | 33.62 | $4.14 \times 10^{-2}$ |
| 1:2 | 1.0000 | 0.67 | $1.26 \times 10^{-1}$ | 49.79 | $6.13 \times 10^{-2}$ |
| 1:4 | 0.9993 | 0.50 | $3.66 \times 10^{-1}$ | 77.32 | $9.52 \times 10^{-2}$ |

Mathematically, the fit parameter, $b$, drives the exponential behavior of our fit. When on the order of magnitude of $10^{-1}$ the
exponential behavior of our fit becomes most exaggerated. This is exemplified in Figure 13, below, for depth ratios 1:4 and 1:2.

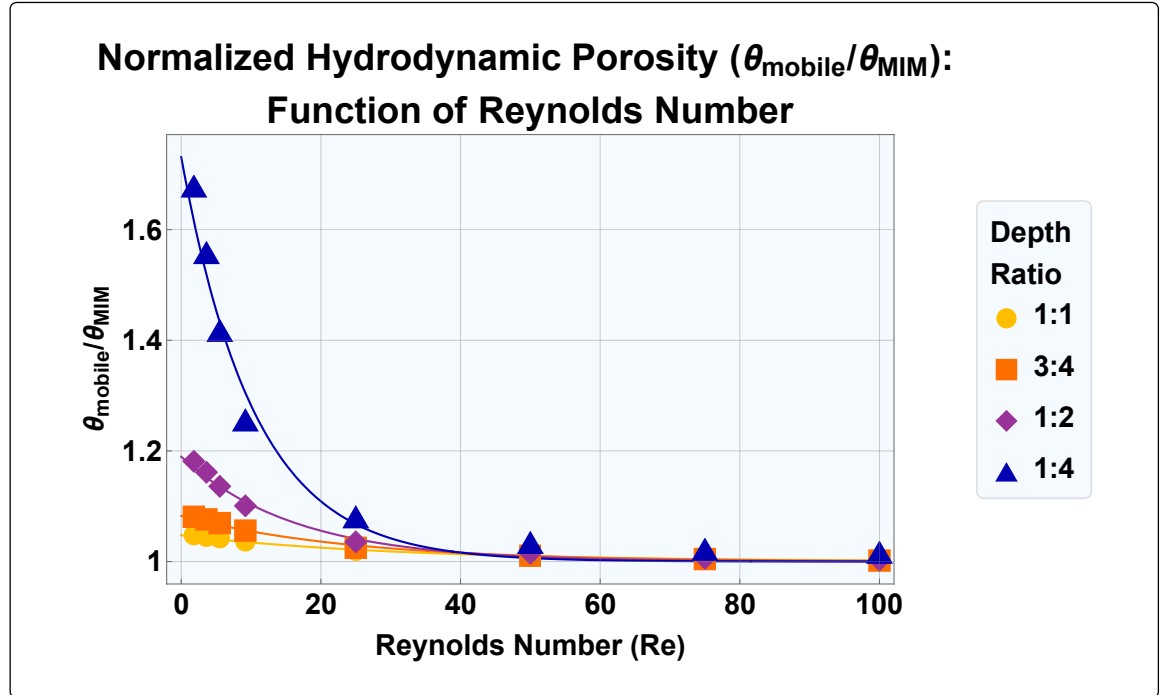

Figure 13: The exponential decay of hydrodynamic porosity, $\theta_{mobile}$, as a function of Reynolds number, $Re$, for media with cavities of varying channel-cavity depth ratios. For ease of comparison, $\theta_{mobile}$ is normalized by the value that corresponds to the Mobile-
Immobile Zone model, $\theta_{MIM}$.





In Figure 13, we normalize the hydrodynamic porosity, $\theta_{mobile}$, by the *static* mobile-zone porosity value, $\theta_{MIM}$, that results from enforcement of the Mobile-Immobile Zone model in the dead-end pore space, defined as:

$$\theta_{MIM} = \xi_{MIM}\,\theta \tag{13}$$

The geometric pore-space partitioning coefficient, $\xi_{MIM}$, is determined by the relative magnitudes of the through-channel and cavity volumes of each dead-end pore geometry. For 1:1 depth ratio pictured in Figure 4, $\xi_{MIM} = 4/5$. For the 1:2 depth ratio pictured in Figure 6, however, $\xi_{MIM} = 8/12$.

## 6 Calculating Pore-Scale Flow Velocity and Hydrodynamic Porosity Parameters

### 6.1 Pore-Scale Flow Velocity

We can use this newfound exponential relationship outlined in Eq. (12) to fill in the details of Eq. (5):

$$v_{pore} = \frac{v}{(a + b\,e^{-c\,v_{pore}})\,\theta} \tag{14}$$

Here, we remind readers that $v$ is the volumetric velocity used in Darcy's or Forchheimer-Dupuit's law (Eq. (2) and (3), respectively). To illustrate the ease at which the exponential relationship in Eq. (12) can be incorporated into current models, we provide a brief example utilizing the idealized pore geometry provided in Figure 4, the associated exponential fit coefficients (i.e., $a = 0.80$, $b = 3.80 \times 10^{-2}$, and $c = 25.91$ s/m), and the properties of water assumed by Kahler and Kabala (2016). In this example, we assume the volumetric velocity has a magnitude $10^{-3}$ (m/s) and a total porosity ($\theta$) of 0.4 – an arbitrary total porosity value in the range typically measured for unconsolidated, unwashed media (i.e., 20 – 45%) (Woessner and Poeter, 2020). Using only 3 Picard iterations with an initial guess of $2 \times 10^{-3}$ (m/s), we converge to 3 decimal places and a pore-scale flow velocity of $2.99 \times 10^{-3}$ (m/s). Below, we illustrate the power of *Mathematica's* built-in *Nest* function, which will, for the same parameter set, converge for an initial guess spanning *twenty orders of magnitude*. In the code provided below, we use the sister, *NestList* function to quickly determine the number of iterations needed for convergence.





```
? NestList
```

| Symbol | ⓘ |
|---|---|
| NestList[*f*, *expr*, *n*] gives a list of the results of applying *f* to *expr* 0 through *n* times. | |
| ⌄ | |

```
vV = 0.001; (*hypothetical volumetric velocity*)
IG = 0.002; (*initial guess*)
vP = NestList[        0.001         &, 0.002, 4] (*pore-scale velocity*)
              ─────────────────────
              n * (a + b * Exp[-c #])

{0.002, 0.00299016, 0.00299344, 0.00299345, 0.00299345}
```

In the table below, we repeat this process for an initial guess 10 orders of magnitude larger and smaller than our original guess
to illustrate the robust nature of this method.

**Table 4: The pore-scale flow velocity (m/s) can be determined in only a few Picard iterations.**

| | Pore-Scale Velocity (m/s) | | | |
|---|---|---|---|---|
| | Iteration Number | | | |
| Initial Guess | 1 | 2 | 3 | 4 |
| $2 \times 10^{-3}$ | 0.00299016 | 0.00299344 | 0.00299345 | 0.00299345 |
| $2 \times 10^{-13}$ | 0.00298329 | 0.00299342 | 0.00299345 | 0.00299345 |
| $2 \times 10^{7}$ | 0.003125 | 0.00299388 | 0.00299346 | 0.00299345 |


From this demonstration, we conclude that implementation of Eq. (12) is no more mathematically burdensome to implement
than the constant porosity model, which results in unnecessary error.

**6.2 Hydrodynamic Porosity Fit Parameters**

In this work, we illustrate how to solve for the exponential fit parameters in Eq. (11) and (12) through numerical simulation.
But this is an idealized case – porous media have an array of randomly distributed and sized pore spaces, not to mention drastic
changes in pore geometry. If we wanted to use the relationship provided in Eq. (12), we would need to quantify the volumetric
and pore-scale flow velocities to then be able to determine the corresponding exponential fit parameters. This could be achieved
through column experiments with a known pressure gradient (to solve for the volumetric flow velocity), and a measurable
tracer pulse (to solve for the pore-scale flow velocity). Given that Eq. (12) contains three unknown parameters, we would need
to conduct this experiment *at least* three times, each at a different, judiciously selected flow condition. To improve fit quality
and reduce the effect of measurement noise, we recommend an overdetermined system in practice.





In this example, we work with an artificial dataset, generated from use of Eq. (14) and the exponential fit parameters we used in our last example (i.e., $a = 0.80$, $b = 3.80 \times 10^{-2}$, and $c = 25.91 \, s/m$). We assume we have measured volumetric
velocities that correspond to the laminar flow regime ($Re = 1$–100); for the characteristic height we assign to the dead-end pore geometry (i.e., 0.001 m), the corresponding pore-scale velocity range is approximately 0.001–0.1 (m/s). Let's assume we measure the volumetric flow velocities 0.01, 0.05, and 0.09 (m/s). Given this assumption, we can calculate the corresponding pore-scale flow velocities:

**Table 5: Artificial volumetric and pore-scale flow velocity (m/s) data set, generated from Eq. (13).**

| Volumetric Velocity (m/s) $\times 10^2$ | Pore-Scale Flow Velocity (m/s) $\times 10^2$ |
|:---:|:---:|
| 1 | 3.05939 |
| 5 | 15.6122 |
| 9 | 28.1241 |


Moving forward, we assume that we have measured these values experimentally and that we actually do not know the values of our exponential fit parameters, *a, b,* and *c*. We can use the synthetic dataset provided in Table 5 to calculate them. This is easily achieved through use of *Mathematica's FindFit* function. Specifying a Newton solution method, we arrive at the anticipated parameter values, *exactly* in 100 iterations (the default value used by the *FindFit* function).

```
? FindFit
```

> **Symbol**                                                                    ⓘ
>
> FindFit[*data, expr, pars, vars*] finds numerical values of the
>
>     parameters *pars* that make *expr* give a best fit to *data* as a function of *vars*.
>
> FindFit[*data, {expr, cons}, pars, vars*] finds a best fit subject to the parameter constraints *cons*.

```
vV = {0.01, 0.05, 0.09};
                vV
vP = Nest[ ─────────────── &, 1, 4];
           n * (a + b * Exp[-c #])
vData = Transpose[{vP, vV}];

Clear[a, b, c]
FindFit[vData, (n * (a + b * Exp[-c v])) * v, {a, b, c}, v, Method → "Newton"]

{a → 0.8, b → 0.0379998, c → 25.9902}
```


Here we note that we have utilized the *arbitrary precision* assigned by *Mathematica* to generate a set of pore-scale velocity values. Realistically, the precision of our measurements would be restricted by our measurement device. Thus, we re-run the above calculation, but this time with only three significant digits rather than the six that we previously used. This time, we arrive at *a, b*, and *c* values of 0.801, 0.0366, and 26.4, respectively. The error in our estimated values is 0.09%, 3.65%, and
1.73%. If we increase the precision of our measurements to four significant digits, the error in our estimated values reduces to





0.14%, 0.10%, and 0.25%, respectively. Clearly, accurate estimation of *a, b*, and *c* will require numerical fine-tuning and surplus velocity data. This process will need to be conducted via numerical and experimental column tests, which we are currently pursuing.

## 7 Discussion

At scale, the implications of these results are easily observed. For example, consider a periodic medium well-approximated by the dead-end pore model with a measurable total porosity of 0.4. If we assume the non-dimensional idealized pore space pictured in Figure 4, we know that the entire area of the through-channel contributes to the mobile porosity of the medium. The immobile zone is occupied by the vortex, which resides within the dead-end pore. For a Reynolds number of 1, the pore-space partitioning coefficient ($\xi$) of the dead-end pore geometry is approximately 0.84. Given this value, we can calculate the

hydrodynamic porosity of the medium using Eq. (10):

$$\theta_{mobile} = \xi\,\theta$$

Resulting, the medium has a hydrodynamic porosity of approximately 0.34. If we had used the boundary-driven flow condition to determine $\theta_{mobile}$ (provided in Figures 4 and 5), we would have under-approximated the value at 0.32 by roughly 5%; similarly, we would have over-approximated the immobile porosity by roughly 20%. For the most exaggerated channel-cavity

depth ratio we test (1:4), use of the boundary-driven model would have resulted in a 42% error in $\theta_{mobile}$. This is due to the deep impingement of the through-flow into the dead-end pore for this configuration. We summarize these calculations in the table below:

**Table 6: Error in the Mobile-Immobile Zone Model for a Periodic Medium with a Total Porosity of 0.4 and cavity geometry pictured in Figure 4.**

|  | Pore-Space Partitioning Coefficient, $\xi = \theta_{mobile}/\theta$ | Mobile Zone Porosity, $\theta_{mobile}$ | Immobile Zone Porosity, $\theta_{immobile}$ |
|---|---|---|---|
| **Hydrodynamic Porosity Model** | 0.84 | 0.336 | 0.064 |
| **Mobile-Immobile Zone Model** | 0.8 | 0.32 | 0.08 |
| **Mobile-Immobile Zone Model Error** (%) | | **5%** | **20%** |


Comparing the shear-driven and boundary-driven models, we see that the error in the latter (in terms of the pore-space partitioning coefficient) is largest for flows approaching the creeping flow regime and does not become less than 1 until a Reynolds number of roughly 49 for the channel-cavity depth ratio of 1:1. For the exaggerated flow geometries (i.e., 3:4, 1:2,



and 1:4 depth ratios), the Reynolds number must exceed 51, 48, and 45, respectively. Referring to Figure 5, we see a 17%
error associated with the use of the boundary-driven model.

The exponential fit quality is unsurprising. Exponentials play an important role in the solution of differential equations and are common to groundwater flow modeling. Consider the use of exponential forms in the various renditions of the well function (e.g., Theis (1935), Hantush (1960), etc.), Gardener's equation for hydraulic conductivity (Gardener, 1958), and even the
relationship between soil water content and electrical resistivity (Pozdnyakov et al., 2006). In nature, an exponential decline in hydraulic conductivity with depth is considered a hallmark of catchment hydrology (Ameli et al., 2016).

Naturally, next steps for this work are in application to groundwater flow models at the macro-scale. The flow geometry pictured in Figure 4 is an oversimplification of the pore space, meaning that results from this work are purely illustrative of
order of magnitude. These results could be made more realistic by establishing the Reynolds number dependence for a more exhaustive set of flow geometries, and more importantly, geometries that exhibit periodicity. The ability to quantify the exponential fit parameters at sites needing remediation, as we previously discussed, is also necessary to demonstrate the ease with which this relationship can be tailored to any given media. We present the results from the former research avenue in our associated work, *Hydrodynamic Porosity: A Paradigm Shift in Flow and Transport Through Porous Media, Part II,* and are
currently pursuing the latter research avenue.

## 8 Study Limitations

The results of this study are not exact. The location of the separatrix is determined via a stream plot with limited resolution. From this stream plot, the separatrix location, and ultimately the hydrodynamic porosity, $\theta_{mobile}$, is approximated by a graphical method that is entirely dependent on user input. However, the repeatability of this measurement process is acceptable
– for the idealized pore space pictured in Figure 4 (i.e., a channel-cavity depth ratio of 1:1), and a Reynolds number of 0.01, the maximum penetration depth of the separatrix was found to vary by 0.005 across 5 measurements. Relative to an average penetration depth of 0.25, this variation accounts for less than 2% of the magnitude of the mean. Like the values reported in Section 6, these errors are purely illustrative of order of magnitude. Both quantities are dependent on the consistency of user input and will also vary between users.


Additionally, we consider only the immobile zones generated by dead-end pores and cavity-like structures that generate flow separation. We do not consider the immobile zones that result from molecular forces between the media and the flowing solution, nor do we consider surface tension effects. However, we assume that these immobile zones, just as the ones studied in this work, would behave similarly so long as the Reynolds number of the interstitial flow is appropriate.





## 9 Conclusions

In this work, we explore a physical phenomenon that has been largely neglected in the literature. Generally, the porosity of a medium that is conducive to through-flow, what we define as *hydrodynamic porosity*, $\theta_{mobile}$, is still thought to be a static parameter. Researchers define this quantity using the cumulative volume of the flow through-channels (i.e., mobile zones); cavities and other effectively immobile zones are not considered to significantly contribute to the through-flow volume. We find this approximation to be a notable oversimplification in the creeping and inertial flow regimes, leading to a significant misunderstanding of induced subsurface flows and the accompanying transport processes that determine the efficacy of groundwater remediation projects.

Although a few researchers have previously acknowledged the dependence of what they analogously define as effective porosity on the flow velocity in rock formations, these researchers did not quantify this relationship. In this work, we demonstrate and quantify, explicitly, the dependence of hydrodynamic porosity, $\theta_{mobile}$, on pore-scale flow velocity, for the first time. We begin with a direct replication of the results provided by Kahler and Kabala (2016), wherein the boundaries between the mobile and immobile zones of porous media are shown to be hydrodynamic in nature. We then study the movement of this boundary over a range of interstitial, or channel-based, Reynolds numbers in the laminar flow regime in porous media. The movement of this boundary defines the pore-space partitioning coefficient, $\xi = \theta_{mobile}/\theta$ (i.e., the fraction of the pore space conducive to through-flow), and therefore the hydrodynamic porosity, $\theta_{mobile}$, of the medium. Given the direct proportionality between these quantities, we find both to have an exponential dependence on Reynolds number or pore-scale flow velocity, which we provide in Eq. (11) and (12). Finally, we show that this dependence can be easily incorporated into porous media flow modeling using only a few Picard iterations, even with an initial guess that is over 10 orders of magnitude off.

The flow-dependent nature of hydrodynamic porosity, $\theta_{mobile}$, plays an unmistakable role in the transport of contaminants into and out of immobile zones and is therefore responsible for contributing to phenomena like contaminant rebound after the completion of active remediation strategies (e.g., steady flow pump-and-treat). Researchers who incorporate this phenomenon into their models could better mitigate these transport phenomena with novel strategies, such as rapidly pulsed pump-and-treat remediation.

**Data and Code Availability**

The simulation data that support the findings of this study, and the corresponding *Mathematica* code files, are available in Open Science Framework at DOI 10.17605/OSF.IO/P2EMN. All files are also provided in pdf format for readers that do not have access to *Mathematica*.



**Author Contributions Statement**

Conceptualization (Z.J.K.), data curation (A.H.Y.), formal analysis (A.H.Y.), investigation (A.H.Y. and. Z.J.K.), methodology (A.H.Y. and. Z.J.K.), project administration (Z.J.K.), supervision (Z.J.K.), validation (A.H.Y.), visualization (A.H.Y.), writing – original draft (A.H.Y. and. Z.J.K.), writing – review & editing (A.H.Y. and. Z.J.K.).

**Declaration Of Competing Interest**

The authors declare no competing conflict of interest.

**Acknowledgements**

This work is, in part, supported by a grant, OPP1173370 (Sanitation Technology Cluster), from the Bill & Melinda Gates Foundation through Duke University's Center for WaSH-AID. All opinions, findings, and conclusions or recommendations

expressed in these works are those of the author(s) and do not necessarily reflect the views of the Foundation, Duke, or the Center.





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
