# Peer review of "Hydrodynamic Porosity: A Paradigm Shift in Flow and Contaminant Transport Through Porous Media, Part I"

_Hydrology and Earth System Sciences, 2023_

## Referee Comment (RC1)

**On the meaning of porosity, hydrodynamic porosity and transport**

Jesus Carrera

GHS, IDAEA, CSIC, Barcelona, Spain

Review of; "Hydrodynamic Porosity: A Paradigm Shift in Flow and Contaminant Transport Through Porous Media, Parts I and II", by August H. Young and Zbigniew J. Kabala

**1. Introduction**

This document contains a review of two papers entitled "Hydrodynamic Porosity: A Paradigm Shift in Flow and Contaminant Transport Through Porous Media" (parts I and II, Young and Kabala, 2023a and b, respectively). In these papers, the authors introduce the concept of hydrodynamic porosity, as the fraction of porous medium connected through flow lines traversing the medium. Further, they find a relatively simple expression for the relationship between hydrodynamic and (total) porosity, as a function of Reynolds number. While this is a relevant contribution, the authors make a number of statements that are, not only ungranted, but also misleading. Together with other contributions by other authors in "fashion" journals, these statements add confusion to the transport field. The spirit of HESS discussions (Copernicus, n.d.) calls for "public commenting on the submitted preprint **by the referees, authors, and other members of the scientific community ... stimulate further deliberation...**". In this spirit, prior to the actual assessment of the papers, I prefer to clarify some basic issues that are generally accepted the scientific community. Therefore, prior to the actual review in Sections 4-8, I discuss basic concepts on porous media (section 2), and (anomalous) transport (section 3).

**2. Porosity, specific surface and permeability are well defined and not transport attributes**

A porous medium consists of one or several fluid phases (unless specified otherwise, I will assume a single fluid, water) and, usually, several solid phases (minerals, biofilms, organic matter, etc.). The space between solid phases is called "void space". Porosity is the ratio of void volume to total volume of a porous medium. Specific surface is the interfacial surface between solid phases and void. Given the multiplicity of phases, the concept of specific surface is also the subject of debate. Note that these definitions simply describe the geometric structure of the medium. They are relevant for all phenomena that may occur in the medium, but in principle, they are independent of those phenomena. They may change through coupling. For example, porosity may change in response to changes in stress, which is key factor in mechanical coupling and in the definition of storage, or in response to dissolution/precipitation, which is one of the motivations of reactive transport. But, on a first approximation, changes are assumed small (or taking place over very long times), so that they are neglected.

Permeability is more complex. In principle, it represents momentum conservation (viscous forces exerted by the medium counter pressure forces exerted by the fluid). Darcy's Law results from the fact that that viscous forces are proportional to velocity. The critical issue here is that, for Darcy's law to be valid, the geometry of flow lines must remain unchanged when the flux is changed, which requires flow to be "slow laminar" (Re<1, where Re is Reynolds Number, though often accepted Re<10). For higher Reynolds numbers, it is usually accepted that the energy loss is more than proportional to the flux. Young and Kabala (2023a) provide a lucid discussion on this topic in Section 4.2.1.

The summary of this section is that the key hydrodynamic parameters of porous media (porosity, specific surface and permeability) are well defined. They are not attributes of any specific phenomenon, but descriptors of the pore geometry. Certainly, using them is not "oversimplistic".

**3. On solute transport and the adjectives of porosity**

The above concepts are not sufficient to simulate complex phenomena, especially because of the heterogeneous nature of geologic materials. Therefore, researchers have been forced to introduce additional parameters. In particular, numerous adjectives have been added to porosity in an effort to improve the accuracy of transport formulations. A small fraction of these adjectives are mentioned by Young and Kabala (2023a) in Section 2.1. So, I agree with them that the term effective porosity would be ambiguous. However, what I want to emphasize here is that the main goal of these alternative definitions is not so much to study the structure of porosity or to distinguish between mobile and immobile water as to describe solute transport. In fact, the original work of Van Genuchten and Wierenga (1976) was motivated by the anomalous transport features they frequently observed in breakthrough curves. While they mentioned unsaturated flow conditions, which enhance the role of low permeability aggregates, their work has been extended to saturated flow conditions by numerous authors. Carrera et al. (2022) review the various alternative formulations that have been produced and that, directly or indirectly, can be considered extensions of the mobile-immobile model. They argue for the need of this model to account for observed chemical localization, which is essential for remediation. Therefore, I do not consider these models to be outdated, but adequate to represent the numerous departures from "normal" (i.e., Fickian) transport.

Dead end pores are one cause of "anomality" (it is ironic that we term "anomalous" what is consistently observed in reality. However, I doubt they are critically relevant. Diffusion dominates at the pore scale, as shown in Table 1, which compares pore advection and diffusion times for a range of water flues (from around 1cm/d to 1 m/d), and pore sizes (form 1 cm to 10 microns). It is clear that, except for gravels, diffusion dominates. That is, transport is diffusion dominated in the regions with largest specific surface, which are the ones most likely to hold contaminants.

Table 1: Reynolds and Peclet Numbers for a range of water fluxes and pore sizes typical of unconsolidated aquifers.

| Flux (m/s) | mean veloc (m/s) | Pore size (m) | Reynolds num vd/nu | adv time (s) | Diff time (s) | Pore Peclet (-) |
|---|---|---|---|---|---|---|
| 1E-07 | 4E-07 | 1E-02 | 1E-03 | 3E+04 | 5E+04 | 2.000 |
| 1E-06 | 4E-06 | 1E-02 | 1E-02 | 3E+03 | 5E+04 | 20.000 |
| 1E-05 | 4E-05 | 1E-02 | 1E-01 | 3E+02 | 5E+04 | 200.000 |
| 1E-07 | 4E-07 | 1E-03 | 1E-04 | 3E+03 | 5E+02 | 0.200 |
| 1E-06 | 4E-06 | 1E-03 | 1E-03 | 3E+02 | 5E+02 | 2.000 |
| 1E-05 | 4E-05 | 1E-03 | 1E-02 | 3E+01 | 5E+02 | 20.000 |
| 1E-07 | 4E-07 | 1E-04 | 1E-05 | 3E+02 | 5E+00 | 0.020 |
| 1E-06 | 4E-06 | 1E-04 | 1E-04 | 3E+01 | 5E+00 | 0.200 |
| 1E-05 | 4E-05 | 1E-04 | 1E-03 | 3E+00 | 5E+00 | 2.000 |
| 1E-07 | 4E-07 | 1E-05 | 1E-06 | 3E+01 | 5E-02 | 0.002 |
| 1E-06 | 4E-06 | 1E-05 | 1E-05 | 3E+00 | 5E-02 | 0.020 |
| 1E-05 | 4E-05 | 1E-05 | 1E-04 | 3E-01 | 5E-02 | 0.200 |

Further insight can be gained from random walk simulations by Bolster et al. (2009) shown in Figure 1, which illustrate that diffusion causes solutes to equilibrate in the recirculating (i.e., closed flowlines) flow regions (if anything, recirculation accelerates equilibrium).

[Figure]

Figure 1: Simulations of Flow through a channel with periodically varying aperture: Flow lines (left) and particles locations resulting from random walk simulations with Pe=1000 (center) and Pe=10 (right) (modified from Bolster et al, 2009)

The ultimate motivation for this long discussion is that what controls the mean arrival is the total porosity. Velocity can be locally very large but diffusion will tend to equilibrate immobile regions at the pore scale, as shown in Figure 1. The key parameter is not so much the mobile porosity as the time it takes for equilibrium. Figure 2 displays breakthrough curves obtained with multi-rate-mass-transfer (multiple immobile zones characterized by a memory function with log-log slope of ½ for varying characteristic diffusion times, Carrera et al., 1998) model for transport along a 9 cm long column. All models are identical except for the mobile-immobile porosities (the total porosity is always 0.4). When the characteristic diffusion time is much smaller than the advection time, the curves are virtually identical. Peak arrival is fast with small mobile porosity only when the characteristic diffusion time is much larger than the advection time. In all cases, the mean arrival time is the same ($t=V_w/Q$), as demonstrated for this kind models by Haggerty and Gorelick (1995) and Carrera et al. (1998).

[Figure]

Figure 2: Simulations breakthrough curves of a pulse injection in arithmetic (above) and log-log (below, to illustrate that fast arrival BTCs display longer tails) varying the mobile porosity (total porosity is always 0.4) and the memory function maximum time. Mean arrival time is the same (1624 min) for all curves.

A last comment, while these models were originally derived for diffusion into immobile zones (soil aggregates, rock matrix blocks, or even intragranular discontinuities, Wood et al., 1990),

similar models have been developed for "slow advection (Berkowitz and Scher, 1995). Unfortunately, the models are mathematically identical (see discussion by Carrera et al., 2022). Medina and Carrera (1996) showed that advective and diffusive models that calibrate equally well under laboratory or field tracer test conditions may lead to radically different results under natural field conditions (typically, much slower mean flux). Carrera et al. (1998) suggested using tracer tests with different fluxes to distinguish advective and diffusive exchange. I emphasize this because, to me, the most notable result of Young and Kabala (2023a and b) is precisely that exchange displays an advective component even in dead-end pores.

**4. Overall assessment of the papers by Young and Kabala (2023a and b)**

The first paper (Young and Kabala, 2023a) can be divided in two parts. The first part is devoted to argue that (1) pump and treat has not been very effective because done under constant flow rate, and (2) porosity is not a single value, but depends on velocity. The second part is devoted to show numerical simulations of laminar flow along a straight pore with a square cavity to the side. Simulations are performed by varying the flow rate (Reynolds Number) and the ratio of pore width to cavity depth. Simulations show a closed circulation area (vortex). The authors then separate the pore space in the area where water flows through (continuous flowlines from inflow to outflow boundaries) and the area where it does not (vortex). The ratio of the first to the total area is termed "hydrodynamic porosity" and its fraction of the total porosity is fitted to a constant plus an exponential of the Reynolds Number.

The second paper (Young and Kabala, 2023b) is an extension of the first one, where the cavities ("dead-end" pores) are extended to other geometries (rectangular, triangular, and semicircular). The resulting hydrodynamic porosity fraction is also fitted to a constant plus an exponential of the Reynolds Number.

The topic of the papers is appropriate for HESS and they are generally well written (see editorial comments below) in the sense that they are understandable. However, the tone is too self-serving, uncritical of their own work and critical of everyone else. Worse, much of the text (basically the 10 first pages) is irrelevant to the actual results (plus questionable, see Section 6 below). Challenging the views and definitions generally accepted by the scientific community is needed and will lead to badly needed "paradigm shifts", but I am afraid that the challenges are poorly argued and the results do not question current views.

In summary, I think that the point the authors try to make is not supported by their results (actually, it is the opposite, see section 5 below). This, together with the questionable 10 introductory pages of paper I, imply that **the papers cannot be published in their current form**. Still, I think the work is of value (I found their figures fascinating) and many discussions lucid. Therefore, I recommend the authors to (1) revise their papers according to the general comments in Section 5 below, (2) revise the first 10 pages of paper I according to the comments in Section 6, (3) clean-up and clarify the text according to the comments in Sections 7 (paper I) and 8 (paper II).

**5. General comments**
1. The authors do not show that transport occurs in the hydrodynamic porosity (only advection does). As shown in Section 3, these immobile zones tend to equilibrate with the hydrodynamic porosity in a very short time (ranging from milliseconds to hours, which is small given the typical residence times). What the authors show is that water in "immobile" dead ends is not really immobile. This is paradoxical, because their results imply that equilibrium will occur faster than predicted by diffusion. This implies an

additional dispersion mechanism, discussed by Bolster et al. (2009). Unfortunately, Young and Kabala (2023a and b) do not discuss the velocity, shear, or curl of their vortices. Therefore, it is hard to ascertain this effect, although I suspect it will be very small for the range of Reynolds numbers studied here.

2. The relationship identified between Re and $\theta_{mob}$ is neither discovered (it is fitted and hardly discussed why) nor exact. For one thing, $\theta_{mob}$ is not just a function of Re (this was my first disappointment). You fix the dependence by fitting $\theta_{mob}$ to a set of Re values, having fixed all other parameters (pore geometry, dead end shape, viscosity).

3. It cannot be considered theoretically based. For Darcy's Law to be valid (see discussion in Section 2), the slope should be zero near the origin. But the slope is maximum at the origin with the proposed expression (In fact, the $\theta_{mob}$ graphs suggest that indeed $\theta_{mob}$ tends to become constant as Re tends to zero).

4. As a result, the fits are good, but not exact. Certainly, the coefficient of determination, $R^2$, is not "approximately 1" (this is stated in the papers abstracts of the two papers!) as clearly seen in Figures 13 of both papers. In fact, simple inspection of the one in paper II suggests a R2 of 0.99, instead of the 0.9999 reported in table. $R^2$ is a rather forgiving parameter. We all use it, but exaggerating it is not appropriate. A $R^2$ of 0.99 to fit 8 points with four parameters is not outstanding (unless the model has a theoretical basis)

5. But the problem is more severe, as it is not clear what is being fitted. At the beginning, $\xi$ is defined as the ratio of $\theta_{mob}$ (wouldn't be more clear $\theta_{hyd}$?) to $\theta$. But it is never used afterwards. Instead Figures 13 display $\theta_{mob}/\theta_{MIM}$. I have failed to understand what $\theta_{MIM}$ is. It is defined in Equation (13) as $\theta_{MIM} = \xi_{MIM}\,\theta$, where $\xi_{MIM}$ is "determined by the relative magnitudes of the through-channel and cavity volumes for each dead-end pore" (determined, how?, certainly, it is not the ratio, because, if so, Eq. 13 would not make much sense. In this context, the statement "For example, using Eq. (2), we find that for the square cavity, $\xi_{MIM} = 4/5$" leaves me perplexed. In summary, I am not sure what is being fitted. This is frustrating for me, as reviewer, but also to potential readers. So, I have been forced to read the papers accepting that "somehow" the hydrodynamic porosity drops as the Reynold number increases.

In summary, I see value in the work done, but the presentation needs to be more realistic and accurate.

**6. Why the introduction is inappropriate in Young and Kabala (2023a)**

I generally agree on the 2 pages discussion on the ubiquity and severity of GW pollution, but it very marginally related to the paper objectives. Instead, it might be more appropriate to review the research community efforts to address solute and reactive transport through porous media.

Lines 90-100: The authors classify as "outdated" what everyone else does even before introducing their concept. And claim that the paradigm shift is related to shifting from "pump and treat with steady flow" to pulsed "pump and treat". It is well known that fluctuating the flow rate in any remediation scheme accelerates remediation (Davidson et al., 2004). But there are numerous explanations for this behavior, ranging from shock waves (Sorek et al., 1992 and 2010) to chaotic mixing, increase in dispersion by transient flow, or ejection by curls in dead end pores, which host pollutants. The latter is well argued by Kahler and Kabala (2016), but it is not addressed at all in this paper. Therefore, it leads to frustration. At first, I thought that this paper was about shock waves. After reading the paper by by Kahler and Kabala (2016), I realized that it was related to transient vortices, only to find that all simulations in both papers are steady-state.

The whole section 2 is devoted to define Effective Porosity as the fraction of the medium devoted to transmit water... at this stage, it is not clear what are the authors referring to. Yet they go on a lengthy criticism of the work by others and an ungranted praise of their own work. I found it amusing, but was frustrated by not really understanding what they are talking about.

Line 207-210: Except for deformable media, porosity is clearly a single scalar value (ratio of voids to total volume) that does not depend on flow. 7 pages into the text and I still do not know what this paper is about (probably something related to porosity). It is true that many adjectives are used with porosity, but you do not need to criticize everyone of them!

Line 213: A very basic concept is that "Darcy/Forchheimer velocity" is not a velocity, but the volumetric water flux. Please, do not introduce a new velocity term here (volumetric velocity?, no one uses this term!)

Line 235: "Given our previous discussion, we know that use of the medium's total porosity is an oversimplification", which discussion? Why oversimplification. Your equation (4) yields the mean velocity regardless. So, it is not any oversimplification. It is just a definition, what may be an oversimplification is its candid use for solute transport. So, I suggest that you define what you mean by "the total volume that is conducive to flow". As a result, Eq. 5 is meaningless at this stage (and we are in page 9).

Line 244: The immobile zone of Van Genuchten (spelling!) and Wierenga (1976) is NOT "defined by isolated volumes of cavities or dead-ended pore space adjacently located to well-connected, mobile regions. They refer to low permeability zones where water velocity is very small. This is especially severe in the unsaturated zone, where water and solutes (the primary goal of their work) can be isolated in highly retentive portions, to be bypassed by fast flows around,.... The good news is that we finally learn what you are talking about!

In summary, please rewrite sections 1-3 into a compact introduction motivating what you are going to do.

**7. Editorial comments on paper I**

Line 10: conducive, conductive, but the statement sounds awkward

Line 18: "Finally, we show that this exponential dependence can be easily solved for pore-scale flow velocity through use of only a few Picard iterations, even with an initial guess that is 10 orders of magnitude off". True, but irrelevant from a transport point of view. Probably not worth mentioning it in the abstract.
Line 25: I do not understand "domestic and global populations". Do you mean "urban and global"?

Line 29: "6.5 trillion liters" probably OK for fashion journals, but not needed for scientific journals.

Figure 8 and flow lines plots. I have found these figures puzzling and fascinating. Usually, flow lines are plotted at equal flow intervals, which is clearly not the case here (but do not change it, the figures would not be as beautiful). Instead, describe the color code. It appears that warm colors indicate higher velocity, but it would be nice to know how much.

The terminology of depth, width, depth into the cavity, normalized depth, etc. is often confusing and, I believe, inconsistent between the two papers (also inconsistent is the fitting description).

Line 546, as discussed earlier, v is not a velocity, but a flux. While the term "Darcy velocity" is widely used, I believe it is confusing in these papers.

The whole section 6.1 is a bit of an overshoot. The fixed point theorem ensures fast convergence of Picard iterations for functions as flat as yours. However, I would not emphasize it too much OK in the text, but not in the abstracts!!), just in case a mathematician looks at it.

The examples in Section 6.2 are very unfortunate. A velocity of 2800 m/s is higher that the velocity of sound. You cannot displace water at those velocities anywhere, much less in a porous medium. Please, revise that, just in case a hydrologist looks at it.

**8. Editorial comments on paper II**

Line 25: Equation 1 is a bit careless. Some terms are not clearly defined ($v_{pore}$?, it is a velocity, but it is not clear which), others are defined twice ($a$?), and $c$ is defined as dimensionless (it should be s/m) and I am utterly confused about the units of d.

Lines 42-44: The last statement of the paragraph is bit mysterious: "Further, researchers can expand...". What one would expect at the end of the introduction is a description of the specific objectives of your work.

Figure 1: I would say that what you display is a "washed" porous medium. Unwashed porous media typically contains lots of fines (power law distribution)

Figure 8 caption: I am not sure what you mean by "landscape orientation". I assume you mean "plan view", but this is a 2D object. Therefore, talking about orientation is confusing.
...

**9. References**

Bolster, D., Dentz, M., & Le Borgne, T. (2009). Solute dispersion in channels with periodically varying apertures. *Physics of Fluids*, *21*(5).

Berkowitz, B., & Scher, H. (1995). On characterization of anomalous dispersion in porous and fractured media. Water Resources Research, 31(6), 1461-1466.

Carrera, J., Sánchez-Vila, X., Benet, I., Medina, A., Galarza, G., Guimerà, J. (1998). On matrix diffusion: formulations, solution methods and qualitative effects. *Hydrogeology Journal*, *6*, 178-190.

Carrera, J., Saaltink, M. W., Soler-Sagarra, J., Wang, J., & Valhondo, C. (2022). Reactive transport: a review of basic concepts with emphasis on biochemical processes. *Energies*, *15*(3), 925.

Davidson, B., Spanos, T., & Zschuppe, R. (2004). Pressure pulse technology: an enhanced fluid flow and delivery mechanism. In *Fourth International Conference on Remediation of Chlorinated and Recalcitrant Compounds, Monterey, CA*.

Haggerty, R., & Gorelick, S. M. (1995). Multiple-rate mass transfer for modeling diffusion and surface reactions in media with pore-scale heterogeneity. *Water Resources Research*, *31*(10), 2383-2400.

Kahler, D. M., & Kabala, Z. J. (2016). Acceleration of groundwater remediation by deep sweeps and vortex ejections induced by rapidly pulsed pumping. Water Resources Research, 52(5), 3930-3940.

Medina, A., & Carrera, J. (1996). Coupled estimation of flow and solute transport parameters. Water Resources Research, 32(10).

Sorek, S., Bear, J., Ben-Dor, G., & Mazor, G. (1992). Shock waves in saturated thermoelastic porous media. *Transport in porous media*, 9, 3-13.

Sorek, S., Ronen, D., & Gitis, V. (2010). Scale-dependent macroscopic balance equations governing transport through porous media: Theory and observations. *Transport in porous media*, *81*, 61-72.

Youtube(n.d.) https://www.youtube.com/watch?app=desktop&v=d3CfXQbX_ag

Copernicus(n.d.)https://www.hydrology-and-earth-system-sciences.net/peer_review/interactive_review_process.html

Van Genuchten, M. T., & Wierenga, P. J. (1976). Mass transfer studies in sorbing porous media I. Analytical solutions. *Soil Science Society of America Journal*, *40*(4), 473-480.

Wood, W. W., Kraemer, T. F., & Hearn Jr, P. P. (1990). Intragranular diffusion: An important mechanism influencing solute transport in clastic aquifers?. *Science*, *247*(4950), 1569-1572.

Young, A. H., & Kabala, Z. J. (2023a). Hydrodynamic Porosity: A Paradigm Shift in Flow and Contaminant Transport Through Porous Media, Part I. *HESS Discussions*

Young, A. H., & Kabala, Z. J. (2023b). Hydrodynamic Porosity: A Paradigm Shift in Flow and Contaminant Transport Through Porous Media, Part II. *HESS Discussions*

---

## Author Comment (AC1)

8 February 2024

Prof. Ing. Nunzio Romano
Editor
HESS Editorial Office
Copernicus Publications

**Re:** Author comments for manuscripts **hess-2023-208** and **hess-2023-209**.

Dear Prof. Ing. Romano,

Thank you for the two insightful reviews and your decision to allow us additional time to formulate our response.

To start, we thank Dr. Carrera for the time and effort allocated to reviewing our work and note our appreciation of the many insightful comments and discussion points. We recognize the need to bring more brevity and clarity to our submissions and are pleased to make these adjustments. We also thank anonymous reviewer 2, who raised important points about our scope of work and provided valid suggestions for future work.

We allocate the remainder of this letter to addressing the concerns raised by Dr. Carrera and anonymous reviewer 2, provided in *blue, italic* font; we move through these comments and concerns in sequential order. Concerning Dr. Carrera's review letter, we note the section and page number associated with each comment. Given the complementary nature of our manuscripts, we choose to address reviewer comments on both manuscripts in a single document.

Again, we would like to thank you and both reviewers for taking the time to read and analyze our manuscripts and, particularly, for making insightful suggestions on how to improve them. We believe that in this letter we have addressed all concerns raised by the two reviewers. In addressing these issues, our revised manuscripts will be more appealing to the readership of HESS.

Thank you for your consideration. We look forward to your reply.

Sincerely,

August Young, PhD, EIT
Recent Graduate

Zbigniew Kabala, PhD, PE
Associate Professor of Civil Engineering

**Re:** *Review of; "Hydrodynamic Porosity: A Paradigm Shift in Flow and Contaminant Transport Through Porous Media, Parts I and II", by August H. Young and Zbigniew J. Kabala* by Dr. Jesus Carrera.

**Paper 1**

1. **Comment** (page 1, section 1)
   *In these papers, the authors introduce the concept of hydrodynamic porosity … While this is a relevant contribution, the authors make a number of statements that are, not only ungranted, but also misleading. Together with other contributions by other authors in "fashion" journals, these statements add confusion to the transport field.*

   We recognize the need to bring more clarity to our work and are grateful for the opportunity to do so. We hope that with the described adjustments, the utility in our research is made clear. This said, we politely request that Dr. Carrera clarify which of our statements are "ungranted" and "misleading" as the goal of this work is to provide a positive contribution to the study of flow and transport in porous media, not to muddy the waters.

2. **Comment** (page 2, section 3)
   *Carrera et al. (2022) review the various alternative formulations that have been produced and that, directly or indirectly, can be considered extensions of the mobile-immobile model. They argue for the need of this model to account for observed chemical localization, which is essential for remediation. Therefore, I do not consider these models to be outdated, but adequate to represent the numerous departures from "normal" (i.e., Fickian) transport.*

   Thank you for prompting us to clarify our language. In calling the Mobile-Immobile Model outdated, we intended to convey the fact that it is well-known that the immobile zone is not actually stagnant, yet, with a few exceptions, it remains to be modeled this way. In addition to this clarification, we recognize that it would also be prudent, perhaps, to expand our discussion of the Mobile-Immobile Model and the many extensions that have been made to it – thank you for providing your article as an appropriate starting point. Regrettably, we had missed it in our literature search as it was published in *Energies* (a journal that does not have hydrology or hydrogeology in its scope). That being said, in our manuscripts, we (by design) consider only the flow of non-reactive tracers. In the spirit of "making things as simple as possible, but not simpler," we thus limit our focus to hydrodynamic effects and leave the study of reactive flows for future papers.

3. **Comment** (page 2, section 3)
   *Dead end pores are one cause of "anomality" (it is ironic that we term "anomalous" what is consistently observed in reality. However, I doubt they are critically relevant.*

   Surely, the relevance of dead-end pores is dependent on the media in question; washed media are unlikely to have poorly-connected pore spaces. However, in the case of unwashed media, such as glacial till or fractured rock, dead-end pores make-up a non-negligible portion of the pore space. Indeed, after a brief literature review, we find that the presence of dead-end pores is prolific in the subsurface. For example, in an undisturbed soil core

studied by Lee et al. (2001), the fraction of immobile water content was found to range from 0.42 to 0.82. In the field, Casey et al. (1997) measured the average fraction to be 0.62. Testing in the 1960's reveals the significance of dead-end pores in reservoir rock: Fatt et al. (1966) estimated a total volume of 20% in limestone and shellstone core samples. Coats and Smith (1964) estimated a volume of roughly 10% in sandstone core samples.

A significant immobile zone volume means that solutes will travel along preferential flow paths (i.e., well-connected pore volumes), generally bypassing volumes of dead-end pores. Jaynes et al. (1995) suggests that models that fail to account for immobile zones and preferential flow paths may predict solute movement to be half as fast as it actually is.

Given the fact that we did not previously cover this point in depth, we understand the inclination to disregard the importance of dead-end pores. We will provide a more thorough discussion of this topic by including the following references (among others) in our revised manuscript:

- Notably, in a recent *Nature Communications* article, Bordoloi et al. (2022) demonstrate how microscopic flow structures impact the macroscopic transport of particles in the pore space, which they characterize as a disordered structure of dead-end pores connected to percolating channels. More specifically, the authors link the tailing they observe in arrival times to the presence of particles initially located in dead-end pores. They note that the movement of particles out of the dead-end pore space is relevant to a "broad range of environmental and medical applications," specifically highlighting soil remediation, drug delivery, and filtration. In providing motivation for their work, the authors explain that "a complete quantitative understanding of the rule played by local flow structures on anomalous transport associated to dead-end pores has remained *overlooked*." They conclude that "the role of microscopic structure and flow on the dispersion of particles and solutes remains *poorly understood*."

- Leismann et al. (1988) discuss the "well-known" phenomena of tailing in the modeling of large-scale propagation processes in the subsurface. The authors explain that this effect is attributed to the persistence of pollutants in *immobile zones*, or what they refer to as the "dead-ends" of the pore space.

- Gao et al. (2009) find that immobile water in dead-end pores not only affects solute transport processes, but plays an important role. To this end, the authors note in their abstract that they found a significant volume of immobile water in the soil column, which resulted in anomalous early and breakthrough and tailing. Later, the authors estimate that nearly 40% of the water in the studied soil column is stagnant.

- Khuzhayorov et al. (2010) conclude that zones of immobile liquid, which are defined as regions with poor transport properties and pores with dead-ends, significantly impact transport in a porous medium.

- Yuan et al. (2021) find that the presence of dead-end pores hinders the ability to efficiently remove NAPLs (non-aqueous phase liquids) from the pore space. They

attribute this inefficiency to the slow rate of mass transfer between the mobile (or what they refer to as swept) and immobile zone.

- Lake (1989) explains that viscous fingering instabilities and the presence of stagnant areas or *dead-end pores* are "two key limiting factors" in the efficiency of miscible displacement processes in porous media. The authors go on to note the relevance of this work to remediation processes, CO2 sequestration, and energy extraction.

- Tangentially related, is the optimization of electrode design; in a recent study on the microstructure of porous battery electrodes, Nguyen et al. (2020) describe the effect of dead-end pores on electrode performance.

Although the prevalence of dead-end pores is media-specific, it is widely agreed that transport into/out of these zones is limited (e.g., see Battat et al. (2019), Shin et al. (2016), etc.). Thus, we find it relevant and important to continue and expand on the discussion started by the aforementioned authors.

4. **Comment:** (page 2, section 3)
*Diffusion dominates at the pore scale, as shown in Table 1, which compares pore advection and diffusion times for a range of water flues (from around 1cm/d to 1 m/d), and pore sizes (form 1 cm to 10 microns). It is clear that, except for gravels, diffusion dominates. That is, transport is diffusion dominated in the regions with largest specific surface, which are the ones most likely to hold contaminants.*

First, we would like to note that our paper is focusing mainly on hydrodynamic effects, not diffusive effects.

Second, if we may, the data provided in Table 1 warrant further discussion. Foremost, it is not clear that diffusion dominates; advective time scales are less than or equivalent to diffusive time scales for half of the entries. We further note that when the diffusive time scale is at least one order of magnitude less than the advective time scale, the Reynolds number is approaching zero in the creeping flow regime. As discussed in our work, we choose to study larger Reynolds numbers because mobile-zone porosity is constant in the creeping flow regime. Another interesting feature of the data in Table 1 is the fact that at a Reynolds number on the order of 1E-03, advective and diffusive time scales are on the same order of magnitude. Bordoloi et al. (2022) explains that diffusion alone cannot model the macroscopic power law tailing observed at this Reynolds number. Instead, it is the pore-scale flow structures within dead-end pores that explain this tailing behavior.

5. **Comment:** (page 3, section 3)
*Further insight can be gained from random walk simulations by Bolster et al. (2009) shown in Figure 1, which illustrate that diffusion causes solutes to equilibrate in the recirculating (i.e., closed flowlines) flow regions (if anything, recirculation accelerates equilibrium).*

We do not argue this point; recirculation does indeed accelerate equilibrium and we discuss this concept in lines 247 – 248 and 288 – 292 and Figure 4.

6. **Comment:** (page 3, section 3)
*The ultimate motivation for this long discussion is that what controls the mean arrival is the total porosity. Velocity can be locally very large but diffusion will tend to equilibrate immobile regions at the pore scale, as shown in Figure 1. The key parameter is not so much the mobile porosity as the time it takes for equilibrium.*

Although it may be the case that in the cited study, mean arrival time may be explained by total porosity, we recognize that there are often two time-scales associated with transport in media with dead-end pore spaces. The so-called "late time scaling" in the break-through curves studied by Bordoloi et al. (2022) is a result of particles initially trapped in dead-end pores. Indeed, Haggerty et al. (2000) note that primary mechanism responsible for long breakthrough tails is mass transfer between mobile and immobile zones. Tailing is an important issue in remediation; sites that initially meet cleanup targets may eventually exceed contaminant thresholds due to mechanisms that slowly release these contaminants into the bulk flow (i.e., back diffusion, sorption, etc.) Maghrebi et al. (2015).

7. **Comment:** (page 3, section 3)
*Figure 2 displays breakthrough curves obtained with multi-rate mass-transfer (multiple immobile zones characterized by a memory function with log-log slope of ó for varying characteristic diffusion times, Carrera et al., 1998) model for transport along a 9 cm long column. All models are identical except for the mobile-immobile porosities (the total porosity is always 0.4). When the characteristic diffusion time is much smaller than the advection time, the curves are virtually identical. Peak arrival is fast with small mobile porosity only when the characteristic diffusion time is much larger than the advection time. In all cases, the mean arrival time is the same (t=V_w/Q), as demonstrated for this kind models by Haggerty and Gorelick (1995) and Carrera et al. (1998).*

We thank Dr. Carrera for pressing us on this issue. To address his concerns, we simulate flow through the dead-end pore geometry with a mobile and immobile separatrix and measure the mean arrival time of particles at the geometry inlet. We find that for each of the tested Reynolds numbers, the mean arrival time is larger for the mobile separatrix. Thus, we demonstrate that a variable mobile zone porosity does indeed affect mean arrival time and happy to provide these results in defense of our argument in our manuscript.

8. **Comment:** (page 4, section 3)
*The most notable result of Young and Kabala (2023a and b) is precisely that exchange displays an advective component even in dead-end pores.*

Although it is true that we reaffirm the results of previous studies (see our response to comment #5, where we discuss the concept of vortex-enhanced diffusion), we also define the hydrodynamic quality of mobile-zone porosity, which we argue is a far more significant contribution to the literature.

9. **Comment:** (page 4, section 4)
*The topic of the papers is appropriate for HESS, and they are generally well written (see editorial comments below) in the sense that they are understandable. However, the tone is*

*too self-serving, uncritical of their own work and critical of everyone else. Worse, much of the text (basically the 10 first pages) is irrelevant to the actual results (plus questionable, see Section 6 below). Challenging the views and definitions generally accepted by the scientific community is needed and will lead to badly needed "paradigm shifts", but I am afraid that the challenges are poorly argued, and the results do not question current views.*

To start, we again would like to note that we appreciate the opportunity to clarify our writing and we hope that our next draft is more well-received.

With that said, we do not understand where this *ad hominem* comment about our "*self-serving, uncritical of (our) own work and critical of everyone else*" comes from and what justifies it. Although we do draw contrast to the previous work of others, it is simply in an effort to motivate our own work, which is markedly different.

10. **Comment:** (page 4, section 4)
*In summary, I think that the point the authors try to make is not supported by their results (actually, it is the opposite, see section 5 below).*

We do not feel that this comment is justified given the overwhelming numerical evidence we provide in our manuscripts.

11. **Comment:** (page 4, section 5)
*1. The authors do not show that transport occurs in the hydrodynamic porosity (only advection does). As shown in Section 3, these immobile zones tend to equilibrate with the hydrodynamic porosity in a very short time (ranging from milliseconds to hours, which is small given the typical residence times). What the authors show is that water in "immobile" dead ends is not really immobile. This is paradoxical, because their results imply that equilibrium will occur faster than predicted by diffusion. This implies an additional dispersion mechanism, discussed by Bolster et al. (2009). Unfortunately, Young and Kabala (2023a and b) do not discuss the velocity, shear, or curl of their vortices. Therefore, it is hard to ascertain this effect, although I suspect it will be very small for the range of Reynolds numbers studied here.*

As the titles indicates, the focus of our manuscripts is *hydrodynamic porosity* (i.e., flow modeling) thus we do not actually show where any transport occurs. To this end, we realize that doing so would bolster our argument that hydrodynamic porosity is in fact an important parameter in mass transport modeling, as discussed in response to comment #7.

12. **Comment:** (page 5, section 5)
*2. The relationship identified between Re and $\theta_{mob}$ is neither discovered (it is fitted and hardly discussed why) nor exact. For one thing, $\theta_{mob}$ is not just a function of Re (this was my first disappointment). You fix the dependence by fitting $\theta_{mob}$ to a set of Re values, having fixed all other parameters (pore geometry, dead end shape, viscosity).*

We recognize the need to test the efficacy of the proposed model, which is the purpose of our second paper, where we test different pore geometries as well as sequential geometries. Given the potential application of the work (e.g., groundwater remediation), we chose to

use the viscosity of water which is a standard modeling assumption. We understand that additional lab column studies, such as those of Kahler and Kabala (2019), are needed as well as field studies and intend to pursue this avenue in the future.

**13. Comment:** (page 5, section 5)

*3. It cannot be considered theoretically based. For Darcy's Law to be valid (see discussion in Section 2), the slope should be zero near the origin. But the slope is maximum at the origin with the proposed expression (In fact, the $\theta_{mob}$ graphs suggest that indeed $\theta_{mob}$ tends to become constant as Re tends to zero).*

For clarification, we do not propose an analytical derivation of $\theta_{mobile}$ and do not claim that the experimentally derived definition is theoretically based. We also note that Darcy's law itself is experimentally-derived (Maghrebi et al., 2015; De Marsily, 2003). With that said, we appreciate your pointing out that the slope of the model does not tend to zero as the Reynolds number goes to zero because there should be no change in mobile zone volume when there is effectively no flow. A cubic fit alleviates this issue, but only locally and there is little physical basis for applying this type of fit. Moving forward, we should emphasize that the presented model does not hold in the creeping flow regime (i.e., Reynolds numbers less than 1).

**14. Comment:** (page 5, section 5)

*4. As a result, the fits are good, but not exact. Certainly, the coefficient of determination, R2, is not "approximately 1" (this is stated in the papers abstracts of the two papers!) as clearly seen in Figures 13 of both papers. In fact, simple inspection of the one in paper II suggests a R2 of 0.99, instead of the 0.9999 reported in table. R2 is a rather forgiving parameter. We all use it, but exaggerating it is not appropriate. A R2 of 0.99 to fit 8 points with four parameters is not outstanding (unless the model has a theoretical basis).*

Thank you for pointing this out; exaggeration of the coefficient of determination was not intentional and we will fix the noted inconsistencies.

**15. Comment:** (page 5, section 5)

*5. But the problem is more severe, as it is not clear what is being fitted. At the beginning, $\zeta$ is defined as the ratio of $\theta_{mob}$ (wouldn't be more clear $\theta_{hyd}$ to $\theta$. But it is never used afterwards. Instead Figures 13 display $\theta_{mob}/\theta_{MIM}$. I have failed to understand what MIM is. It is defined in Equation (13) as $\theta_{MIM} = \zeta\theta_{MIM}$, where $\zeta_{MIM}$ is "determined by the relative magnitudes of the through-channel and cavity volumes for each dead-end pore" (determined, how?, certainly, it is not the ratio, because, if so, Eq. 13 would not make much sense. In this context, the statement "For example, using Eq. (2), we find that for the square cavity, $\theta_{MIM} = 4/5$" leaves me perplexed. In summary, I am not sure what is being fitted. This is frustrating for me, as reviewer, but also to potential readers. So, I have been forced to read the papers accepting that "somehow" the hydrodynamic porosity drops as the Reynold number increases.*

Clearly, we moved too fast in describing these equations. We will be sure to improve the clarity of our writing in this section.

16. **Comment:** (page 5, section 5)
*In summary, I see value in the work done, but the presentation needs to be more realistic and accurate.*

We appreciate the recognition and thank you for your comments. The final draft of the paper will be clear and compelling.

17. **Comment:** (page 5, section 6)
*I generally agree on the 2 pages discussion on the ubiquity and severity of GW pollution, but it very marginally related to the paper objectives. Instead, it might be more appropriate to review the research community efforts to address solute and reactive transport through porous media.*

It was our aim to adequately motivate our work by discussing the severity of groundwater pollution and the recent shift toward hands-off remediation strategies. However, given the way in which our introduction was received, it is clear that we need to sharpen the focus of our introduction and more clearly state our research motivation. We thank Dr. Carrera for his suggestion on how to do so.

18. **Comment:** (page 5, section 6)
*It is well known that fluctuating the flow rate in any remediation scheme accelerates remediation (Davidson et al., 2004). But there are numerous explanations for this behavior, ranging from shock waves (Sorek et al., 1992 and 2010) to chaotic mixing, increase in dispersion by transient flow, or ejection by curls in dead end pores, which host pollutants. The latter is well argued by Kahler and Kabala (2016), but it is not addressed at all in this paper. Therefore, it leads to frustration. At first, I thought that this paper was about shock waves. After reading the paper by by Kahler and Kabala (2016), I realized that it was related to transient vortices, only to find that all simulations in both papers are steady state.*

We understand how steady-state simulations preceded by discussion of rapidly-pulsed pump and treat is confusing and misleading to readers. Again, we intend to narrow the focus of the introduction.

19. **Comment:** (page 6, section 6)
*The whole section 2 is devoted to define Effective Porosity as the fraction of the medium devoted to transmit water… at this stage, it is not clear what are the authors referring to. Yet they go on a lengthy criticism of the work by others and an ungranted praise of their own work. I found it amusing, but was frustrated by not really understanding what they are talking about.*

Please see our response to comments #1 and #9.

20. **Comment:** (page 6, section 6)

*Line 207-210: Except for deformable media, porosity is clearly a single scalar value (ratio of voids to total volume) that does not depend on flow. 7 pages into the text and I still do not know what this paper is about (probably something related to porosity). It is true that many adjectives are used with porosity, but you do not need to criticize everyone of them!*

In lines 133 – 150, we discuss the many definitions of porosity used in the literature. Not only is it not just a *single* value (unless we are referring to *total* porosity), it is generally adapted to adequately describe transport. Again, we refer to the work of Bordoloi et al. (2022). In the section entitled *Structure Induced Vortices and Pore-Scale Transport* the authors explain how media with dead-end pores are often modeled as dual media to account for the difference in transport characteristics in well and poorly-connected pore spaces. Surely, this distinction would not be necessary if total porosity were sufficient in describing the nuances associated with macroscopic transport.

**21. Comment:** (page 6, section 6)
*Line 213: A very basic concept is that "Darcy/Forchheimer velocity" is not a velocity, but the volumetric water flux. Please, do not introduce a new velocity term here (volumetric velocity?, no one uses this term!)*

Perhaps this terminology is indeed awkward – but it does indeed exist in the literature. Technically it is indeed a flux, but it has the dimensions of velocity, and, most importantly, it is referred to in the literature as "velocity." Although we to share Dr. Carrera's frustration with the inconsistent use of the language in our field, we are afraid that use of this term has already been well-established.

**22. Comment:** (page 6, section 6)
*Line 235: "Given our previous discussion, we know that use of the medium's total porosity is an oversimplification", which discussion? Why oversimplification. Your equation (4) yields the mean velocity regardless. So, it is not any oversimplification. It is just a definition, what may be an oversimplification is its candid use for solute transport. So, I suggest that you define what you mean by "the total volume that is conducive to flow". As a result, Eq. 5 is meaningless at this stage (and we are in page 9).*

Eq. 5 is mathematical motivation to determine the relationship of mobile-zone porosity on pore-scale velocity. If mobile-zone porosity were a static parameter with no dependence on flow conditions, then Eq. 5 would indeed be meaningless and Eq. 4 would suffice. We further note that the volume conducive to flow *is* defined as the mobile zone volume.

**23. Comment:** (page 6, section 6)
*Line 244: The immobile zone of Van Genuchten (spelling!) and Wierenga (1976) is NOT "defined by isolated volumes of cavities or dead-ended pore space adjacently located to well-connected, mobile regions. They refer to low permeability zones where water velocity is very small. This is especially severe in the unsaturated zone, where water and solutes (the primary goal of their work) can be isolated in highly retentive portions, to be bypassed by fast flows around,…. The good news is that we finally learn what you are talking about!*

While we agree that we could improve the language used in the quoted excerpt to include the existence of immobile zones completely bound by soil or air, or a combination of, we note that our statement is indeed supported by the cited work. Below, we reproduce Figure 1 from Vangenuchten and Wierenga (1976). In red, we highlight volumes of immobile water adjacently located to mobile zones that can be approximated with the dead-end pore model. Those not highlighted are the aforementioned immobile zones completely bounded by soil or air that we neglected to previously acknowledge.

[Figure]

Fig. 1—Schematic diagram of unsaturated aggregated porous medium. *(A)* Actual model. *(B)* Simplified model. The shading patterns in *A* and *B* represent the same regions.

As an aside, Web of Science spells Rien Van Genuchten's name as "vanGenuchten" to avoid confusing "Van" for a second name. Thank you for pointing it out—we will follow their "misspelling" of this name for consistency with this database.

**24. Comment:** (page 6, section 7)
*Line 18: "Finally, we show that this exponential dependence can be easily solved for pore-scale flow velocity through use of only a few Picard iterations, even with an initial guess that is 10 orders of magnitude off". True, but irrelevant from a transport point of view. Probably not worth mentioning it in the abstract.*

We thank Dr. Carrera for his opinion. However, we believe that the issue of Picard iterations may be of interest to readers who do not have formal training in numerical methods or pure math (i.e., hydrologists and geologists).

25. **Comment:** (page 6, section 7)
*Line 25: I do not understand "domestic and global populations". Do you mean "urban and global"?*

We mean "domestic" as in "not foreign." With that being said, we appreciate you flagging the fact that this terminology may be confusing for readers.

26. **Comment:** (page 6, section 7)
*Line 29: "6.5 trillion liters" probably OK for fashion journals, but not needed for scientific journals.*

We present this figure purely to illustrate the magnitude of groundwater pollution in the United States.

27. **Comment:** (page 6, section 7)
*Figure 8 and flow lines plots. I have found these figures puzzling and fascinating. Usually, flow lines are plotted at equal flow intervals, which is clearly not the case here (but do not change it, the figures would not be as beautiful). Instead, describe the color code. It appears that warm colors indicate higher velocity, but it would be nice to know how much.*

We can certainly include velocity scales in our stream plots. Because the focus of the stream plots is to highlight the location of the separatrix, we had originally left these out to so as to not crowd the figures.

28. **Comment:** (page 6, section 7)
*The terminology of depth, width, depth into the cavity, normalized depth, etc. is often confusing and, I believe, inconsistent between the two papers (also inconsistent is the fitting description).*

Thank you for flagging this. We will review both submissions for inconsistencies.

29. **Comment:** (page 7, section 7)
*Line 546, as discussed earlier, v is not a velocity, but a flux. While the term "Darcy velocity" is widely used, I believe it is confusing in these papers.*

Throughout the paper we highlight the fact that we present a two-dimensional analysis. To that end, flow rate is commonly represented as velocity when moving from 3D to 2D.

30. **Comment:** (page 7, section 7)
*The whole section 6.1 is a bit of an overshoot. The fixed point theorem ensures fast convergence of Picard iterations for functions as flat as yours. However, I would not emphasize it too much OK in the text, but not in the abstracts!!), just in case a mathematician looks at it.*

We thank Dr. Carrera for his opinion and note that we have addressed this issue in our response to Comment #24.

31. **Comment:** (page 7, section 7)

*The examples in Section 6.2 are very unfortunate. A velocity of 2800 m/s is higher that the velocity of sound. You cannot displace water at those velocities anywhere, much less in a porous medium. Please, revise that, just in case a hydrologist looks at it.*

Thank you for catching this, we neglected to put a negative in the exponent in the first row of both entries in Table 5. That being said, the calculations used to produce the values in Table 5 are without mistake.

**Paper 2**

32. **Comment:** (page 7, section 8)

*Line 25: Equation 1 is a bit careless. Some terms are not clearly defined ($v_{pore}$?, it is a velocity, but it is not clear which), others are defined twice (a?), and c is defined as dimensionless (it should be s/m) and I am utterly confused about the units of d.*

Thank you for bringing this to our attention; we mixed up the definitions of parameters *c* and *d* in text, as *c* should clearly have the inverse units of velocity and *d* should be dimensionless. Further, to improve the introduction of Equation 1, we recognize the need to preface it with the discussion of relevant parameters (i.e., pore-space volume, through-channel height, etc.).

33. **Comment:** (page 7, section 8)

*Lines 42-44: The last statement of the paragraph is bit mysterious: "Further, researchers can expand…". What one would expect at the end of the introduction is a description of the specific objectives of your work.*

We agree, thank you for bringing this to our attention.

34. **Comment:** (page 7, section 8)

*Figure 1: I would say that what you display is a "washed" porous medium. Unwashed porous media typically contains lots of fines (power law distribution).*

Prior to the introduction of Figure 1, we define an unwashed porous medium as glacial deposits, fractured rock, and filtration media such as granulated activated carbon *meaning those without smoothed surfaces (i.e., spheres)*. However, we neglect to explicitly state this fact and note the necessity of this inclusion. We would like to further clarify that the point of Figure 1 is to highlight the variation in cavity geometry, and not grain size distribution, given that this is the focus of paper two.

35. **Comment:** (page 7, section 8)

*Figure 8 caption: I am not sure what you mean by "landscape orientation". I assume you mean "plan view", but this is a 2D object. Therefore, talking about orientation is confusing.*

Please note that this word choice is standard in Microsoft Office when referring to page orientation (see: File > Page Setup… > Orientation) and therefore should be familiar to readers. We added this clarification to the figure label to instruct readers to view the image such that the page is in landscape orientation rather than portrait.

**Re:** Anonymous reviewer 2 comments

**Paper 1**

1. **Comment:**
   *The paper is very well written with an extended (but not exhaustive) state of the art on the concept of porosity.*

   Thank you for your positive evaluation.

2. **Comment:**
   *The studied pore geometry is oversimplified (channel + rectangular cavity, see fig. 6) and the parameters of the empirical function are fitted for each geometry.*

   While we concede that the pore space is overly simplified by the chosen geometry, we note the prevalent use of this geometry in the study of flows past cavities in real media (e.g., Bordoloi et al. (2022),Kahler and Kabala (2016) ,Battat et al. (2019), etc.) and the use of similarly simplified geometries (e.g., Coats and Smith (1964), Fathaddin et al. (2008), etc.). Although simplistic, the rectangular cavity space is a sufficient first-order approximation of the pore space.

3. **Comment:**
   *As for the first paper, the main missing elements are the 3D geometry of the pores, the change in the pore diameter and the effects of interconnections.*

   Aside from a three-dimensional geometry, the latter two concerns are addressed in our second paper. We agree that expanding the analysis to three dimensions would be complementary and strengthen our two-dimensional simulation results. However, we feel that this work would be best suited for a follow-up paper given the nuances associated with three-dimensional simulation.

**Paper 2**

4. **Comment:**
   *The second paper is an extension of the previous one to different type of cavities (triangle, circular, periodic squares). Again, the pore geometry is oversimplified and the results cannot be extended to realistic porous network. As for the first paper, the main missing elements are the 3D geometry of the pores, the change in the pore diameter and the effects of interconnections.*

   We acknowledge the need to run column testing on washed and unwashed granular media to demonstrate applicability to a real pore network. We also acknowledge the need to expand the analysis to a 3D geometry, but again we note that the 2D geometry, like the rectangular cavity geometry, is a natural starting point for the analysis.

5. **Comment:**
   *Both papers are technically very sound but they could be partly improved by studying the*

*relationship between the fitted parameters and the characteristics of the geometry for example.*

Indeed, this would be an interesting analysis and we thank the reviewer for this suggestion.

6. **Comment:**
*Therefore, these results are of limited interest for a publication in HESS, which promotes research in Earth Systems. However, I leave the final decision to the editor concerning the suitability of both papers for HESS. If yes, I recommend major revision by merging both papers in one manuscript.*

Although we spend much of our time focusing on the pore-scale flow structures in porous media, we emphasize the relevance of our work to hydrological sciences, specifically groundwater hydrology and remediation processes. However, given the prevalence of porous media in nature and engineered systems, we also note the relevance of our work to other journal subject areas (e.g., engineering hydrology, urban hydrology, and water resources management). Regrettably, we must not have made these connections clear enough and plan to do so in our revised manuscript.

**References**

Battat, S., Ault, J. T., Shin, S., Khodaparast, S., and Stone, H. A.: Particle entrainment in dead-end pores by diffusiophoresis, Soft Matter, 15, 3879-3885, 10.1039/C9SM00427K, 2019.

Bordoloi, A. D., Scheidweiler, D., Dentz, M., Bouabdellaoui, M., Abbarchi, M., and de Anna, P.: Structure induced laminar vortices control anomalous dispersion in porous media, Nature Communications, 13, 3820, 10.1038/s41467-022-31552-5, 2022.

Casey, F. X. M., Horton, R., Logsdon, S. D., and Jaynes, D. B.: Immobile Water Content and Mass Exchange Coefficient of a Field Soil, Soil Sci. Soc. Am. J., 61, 1030-1036, 10.2136/sssaj1997.03615995006100040006x, 1997.

Coats, K. H. and Smith, B. D.: DEAD-END PORE VOLUME AND DISPERSION IN POROUS MEDIA, Soc. Petrol. Eng. J., 4, 73-84, 10.2118/647-pa, 1964.

de Marsily, G.: Stochastic Description of Flow in Porous Media, in: Encyclopedia of Physical Science and Technology (Third Edition), edited by: Meyers, R. A., Academic Press, New York, 95-104, 10.1016/B0-12-227410-5/00937-6, 2003.

Fathaddin, M., Awang, M., and Ardjani, K.: A numerical study of pressure changes in dead-end pores, Journal of Hydrology and Hydromechanics, 56, 23-33, 2008.

Fatt, I., Maleki, M., and Upadhyay, R. N.: Detection and Estimation of Dead-End Pore Volume in Reservoir ROCK by Conventional Laboratory Tests, Soc. Petrol. Eng. J., 6, 206-212, 10.2118/1441-pa, 1966.

Gao, G., Feng, S., Zhan, H., Huang, G., and Mao, X.: Evaluation of Anomalous Solute Transport in a Large Heterogeneous Soil Column with Mobile-Immobile Model, Journal of Hydrologic Engineering, 14, 966-974, doi:10.1061/(ASCE)HE.1943-5584.0000071, 2009.

Haggerty, R., McKenna, S. A., and Meigs, L. C.: On the late-time behavior of tracer test breakthrough curves, Water Resour. Res., 36, 3467-3479, 10.1029/2000WR900214, 2000.

Jaynes, D. B., Logsdon, S. D., and Horton, R.: Field Method for Measuring Mobile/Immobile Water Content and Solute Transfer Rate Coefficient, Soil Sci. Soc. Am. J., 59, 352-356, 10.2136/sssaj1995.03615995005900020012x, 1995.

Kahler, D. M. and Kabala, Z. J.: Acceleration of groundwater remediation by deep sweeps and vortex ejections induced by rapidly pulsed pumping, Water Resour. Res., 52, 3930-3940, 10.1002/2015wr017157, 2016.

Khuzhayorov, B. K., Makhmudov, Z. M., and Zikiryaev, S. K.: Substance transfer in a porous medium saturated with mobile and immobile liquids, Journal of Engineering Physics and Thermophysics, 83, 263-270, 10.1007/s10891-010-0341-3, 2010.

Lake, L. W.: Enhanced oil recovery, Prentice Hall Englewood Cliffs, N.J., Englewood Cliffs, N.J.1989.

Lee, J., Horton, R., Noborio, K., and Jaynes, D. B.: Characterization of preferential flow in undisturbed, structured soil columns using a vertical TDR probe, J Contam Hydrol, 51, 131-144, 10.1016/s0169-7722(01)00131-0, 2001.

Leismann, H. M., Herding, B., and Krenn, V.: A Quick Algorithm for the Dead-End Pore Concept for Modeling Large-Scale Propagation Processes in Groundwater, in: Developments in Water Science, edited by: Celia, M. A., Ferrand, L. A., Brebbia, C. A., Gray, W. G., and Pinder, G. F., Elsevier, 275-280, 10.1016/S0167-5648(08)70101-1, 1988.

Maghrebi, M., Jankovic, I., Weissmann, G. S., Matott, L. S., Allen-King, R. M., and Rabideau, A. J.: Contaminant tailing in highly heterogeneous porous formations: Sensitivity on model selection and material properties, J. Hydrol., 531, 149-160, 10.1016/j.jhydrol.2015.07.015, 2015.

Nguyen, T.-T., Demortière, A., Fleutot, B., Delobel, B., Delacourt, C., and Cooper, S. J.: The electrode tortuosity factor: why the conventional tortuosity factor is not well suited for quantifying transport in porous Li-ion battery electrodes and what to use instead, npj Computational Materials, 6, 123, 10.1038/s41524-020-00386-4, 2020.

Shin, S., Um, E., Sabass, B., Ault, J. T., Rahimi, M., Warren, P. B., and Stone, H. A.: Size-dependent control of colloid transport via solute gradients in dead-end channels, Proceedings of the National Academy of Sciences, 113, 257-261, doi:10.1073/pnas.1511484112, 2016.

vanGenuchten, M. T. and Wierenga, P. J.: MASS-TRANSFER STUDIES IN SORBING POROUS-MEDIA .1. ANALYTICAL SOLUTIONS, Soil Sci. Soc. Am. J., 40, 473-480, 10.2136/sssaj1976.03615995004000040011x, 1976.

Yuan, Q., Ma, Z., Wang, J., and Zhou, X.: Influences of Dead-End Pores in Porous Media on Viscous Fingering Instabilities and Cleanup of NAPLs in Miscible Displacements, Water Resour. Res., 57, e2021WR030594, 10.1029/2021WR030594, 2021.